# Offshore freshened groundwater in the Pearl River estuary and shelf as a significant water resource

Chong Sheng [1], Jiu Jimmy Jiao [1,2,3] ✉, Xin Luo [1,2,3], Jinchao Zuo [2], Lei Jia [4] & Jinghe Cao [5]

Large-river deltaic estuaries and adjacent continental shelves have experienced multiple phases of transgressions and regressions to form interlayered aquifer-aquitard systems and are expected to host vast paleo-terrestrial groundwater hundreds of kilometres offshore. Here, we used offshore hydrogeology, marine geophysical reflections, porewater geochemistry, and paleo-hydrogeological models, and identified a previously unknown offshore freshened groundwater body with a static volume up to $575.6 \pm 44.9$ km$^3$ in the Pearl River Estuary and adjacent continental shelf, with the freshwater extending as far as 55 km offshore. An integrated analysis of stable isotopic compositions and water quality indices reveals the meteoric origins of such freshened groundwater and its significance as potential potable water or raw water source for desalination. Hotspots of offshore freshened groundwater in large-river deltaic estuaries and adjacent continental shelves, likely a global phenomenon, have a great potential for exploitable water resources in highly urbanized coastal areas suffering from freshwater shortage.

Many coastal megacities are facing prominent water shortages due to densified population and water contamination, and these problems are believed to worsen under changing climate. Seeking alternative freshwater resources is very important to deal with the increasing freshwater demand worldwide[1,2]. Offshore freshened groundwater (OFG) is the water stored in the pores of sediments and fractures of rocks in the sub-seafloor that has a total dissolved solids (TDS) concentration below that of the overlaying seawater[3]. For most coastal cities that rely heavily on desalinization as the main domestic water supply, the costs of this process remain considerably high if merely using seawater as the raw water[4–6]. The utilization of OFG would enhance the resilience of coastal societies to increased water demand during periods of intense droughts. In some coastal areas, OFG has been already inadvertently exploited by the onshore pumping[7,8]. The available evidence suggests the global

volume of OFG is on the order of $10^5$–$10^6$ km$^3$, which is about two to three orders of magnitude larger than the volume of groundwater extracted globally from continental aquifers since 1900 ($\sim 4.5 \times 10^3$ km$^3$) (Fig. 1a)[3,4], and roughly 5–10% of the total storage of fresh groundwater worldwide (estimated at $13 \times 10^6$ km$^3$[9] and $21.8 \times 10^6$ km$^3$[10]). Although OFG bodies may serve as a potential freshwater supply in the future, numerous scientific gaps in knowledge remain that preclude us from currently exploiting OFG as a sustainable source of freshwater[3]. The formation and evolution mechanisms of OFG remain poorly understood, and many first-order questions related to geometry, provinces, flow dynamics, water origin, and relevant engineering problems need to be urgently addressed[7,11]. This mainly arises from a paucity of appropriate offshore hydrogeological data, in particular the limited coverage of sub-seafloor borehole data[12]. Direct observation of offshore

[1]Department of Earth Sciences, The University of Hong Kong, Hong Kong, China. [2]The University of Hong Kong, Shenzhen Institution of Research and Innovation (SIRI), Shenzhen, China. [3]Southern Marine Science and Engineering Guangdong Laboratory (Zhuhai), Zhuhai, China. [4]Guangzhou Marine Geological Survey, China Geological Survey, Guangzhou, China. [5]Key Laboratory of Ocean and Marginal Sea Geology, South China Sea Institute of Oceanology, Innovation Academy of South China Sea Ecology and Environmental Engineering, Chinese Academy of Sciences, Guangzhou, China. ✉e-mail: jjiao@hku.hk

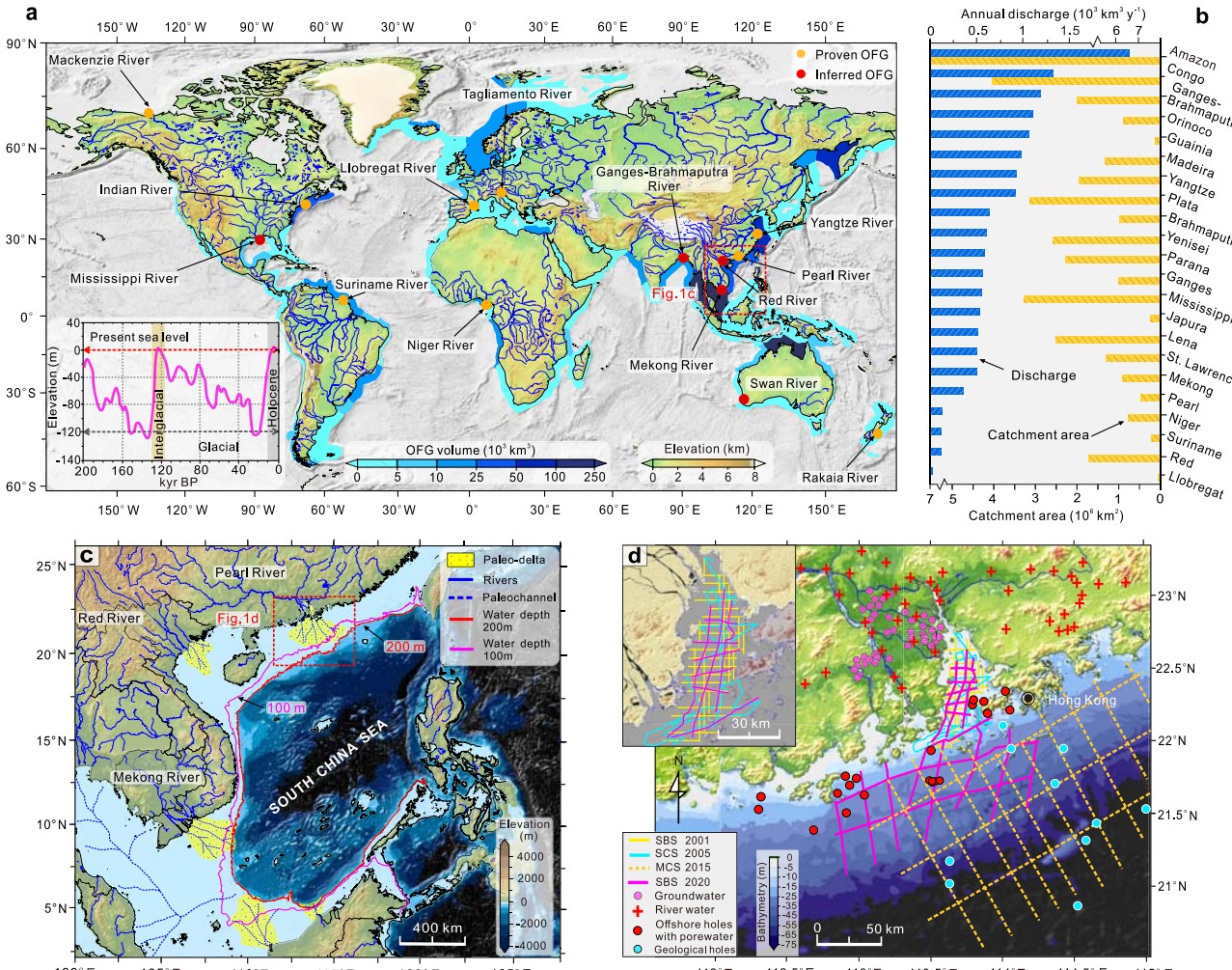

**Fig. 1 | Global map of estimated offshore freshened groundwater (OFG) static volumes in the continental shelves and regional setting of the study area.** **a** OFG static volumes stored in the continental shelves worldwide as estimated by 2D numerical tools[26]. We show locations in the large-river delta-front estuaries (LDEs) and its adjacent continental shelves where the OFG has been demonstrated by direct observational data (yellow points) or inferred by numerical modelling or onshore indicators (red points) (more information can be found in Table S3, supplementary materials). The global eustatic sea-level curve during the past 200 kyr is also inserted here[19]. **b** The average annual discharge and catchment area for the world's largest river. **c** The main larger rivers and subaqueous deltas developed in the LDEs on the continental shelf of the South China Sea. The Pearl River is the largest river in southern China with an annual discharge of $3.6 \times 10^{11}$ m³ to the South China Sea. The buried paleochannel systems were widely developed in the subaqueous deltas[27]. **d** The locations of onshore sampling (groundwater and river water) sites, offshore boreholes, and different types of marine seismic profiles (sub-bottom seismic (SBS), single-channel seismic (SCS), multi-channel seismic (MCS) profiles) obtained in this study.

groundwater reservoir structure and geochemical analyses of porewater data remain very sparse on a global scale[13,14].

Approximately 87% of Earth's land surface is connected to oceans by rivers. By 2025, an estimated 75% of the world's population is expected to live in the area from the shoreline to an elevation of 200 m on land, with many of the remaining 25% living near major rivers[15,16]. Large-river delta-front estuaries (LDEs) as the natural "recorders" of global environmental change represent vital interfaces between continents and oceans[17]. Generally, LDE usually covers subaerial and subaqueous delta systems, including inland areas such as deltaic plains, lowland floodplains, and offshore areas that may extend to the adjacent continental shelf[17,18]. During the Quaternary period, sea level significantly fluctuated near LDEs (see left-bottom subplot in Fig. 1a), and transgression and regression occurred periodically in this period[19]. Therefore, during delta-front progradation, sedimentation is dominated by coarse-grained fluvial deposits, and the river networks will extend further to the sea, whereas during transgressions, fine-grained marine sediments, dominated by clay, silt, or fine sand are deposited[20]. From a

hydrogeological perspective, this geologic scenario leads to the formation of multi-aquifer-aquitard systems in current continental shelves, with high-permeability alluvial and fluvial deposits forming aquifers and low-permeability marine and fluvial over-bank deposits forming interlayered aquitards[21]. Fluvial paleochannels that are usually infilled with high-permeability sediments also act as preferential pathways and have a hydraulic connection to onshore freshwater aquifers, facilitating the further extension of fresh groundwater offshore[22]. Preliminary studies speculate that the OFG may be widely stored in the LDEs and their adjacent continental shelves (Fig. 1a, b), such as those associated with the Yangtze River[4,23], Pearl River[6], Mekong River[21], and Niger River[24]. For example, two offshore pumping tests were conducted in the Yangtze River estuary at an offshore distance of 45 km; the pumping rates were 30.7 and 119.3 m³ h⁻¹ and yielded water with a TDS of 1.277 and 8.131 g L⁻¹, respectively[23], suggesting useful freshwater aquifers may be present in the Yangtze River Estuary and its adjacent shelf. Building on these clues, we hypothesize that the LDEs to be the hotspots of OFG on a global scale. Meanwhile, an in-depth

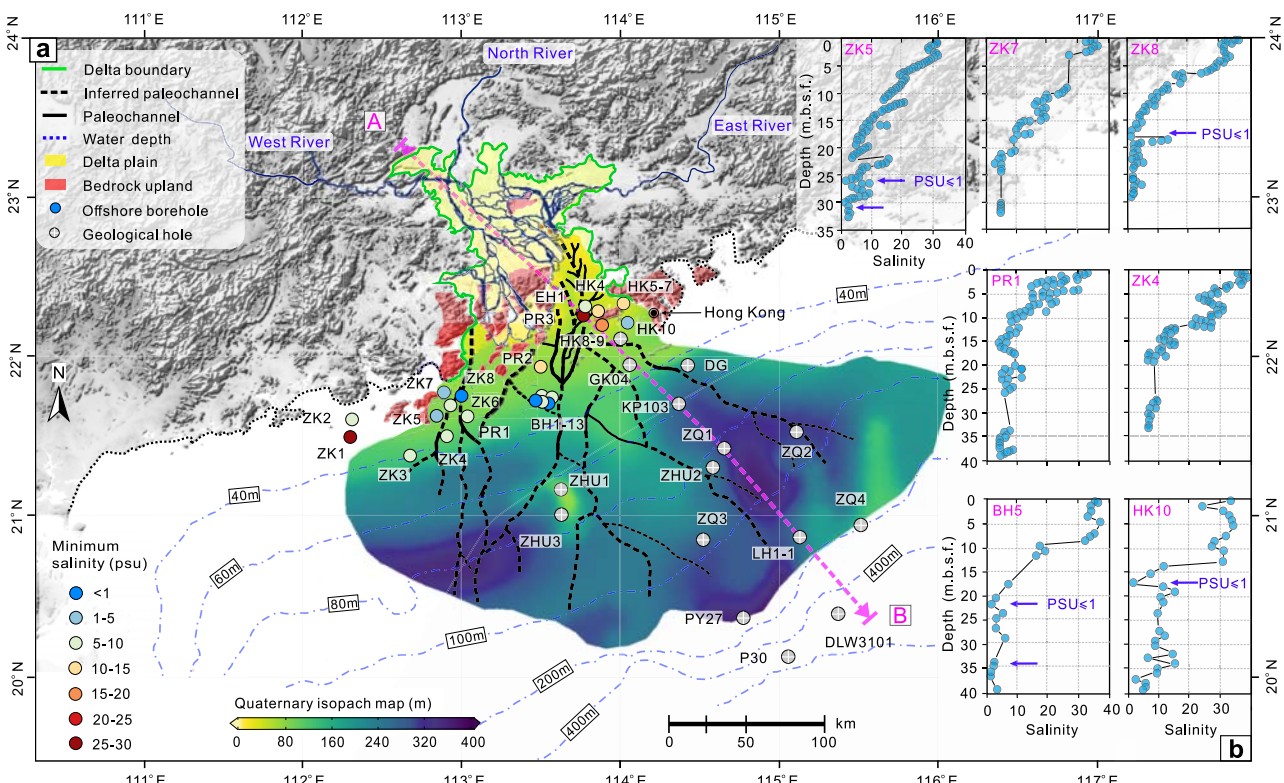

**Fig. 2 | Reconstructed Quaternary isopach map and paleochannels of the subaqueous delta and measured salinity profiles of offshore boreholes in the Pearl River estuary (PRE) and adjacent continental shelf. a** Reconstructed Quaternary isopach map and paleochannels based on offshore borehole logs and high-resolution marine single-channel and sub-bottom seismic profiles on the basis of previous studies[28,33,36,37]. **b** Salinity profiles of offshore boreholes, BH5 and HK10 were drilled for sea sand exploration and only limited cores were sampled for porewater. Data were missing in some sections of other boreholes because of low recovery rates in these sections with coarse materials. The measured salinity profiles of other offshore boreholes in the PRE and adjacent continental shelf can be found in Fig. S2 in the supplementary materials.

understanding of the formation and evolution mechanisms of OFG in the LDEs still lack, preventing effective exploitation of these readily accessible OFG as alternative water sources.

The South China Sea is a marginal sea of the Western Pacific Ocean with an area of $3.5 \times 10^6$ km$^2$, and the buried paleochannel systems are widely distributed in the broader continental shelves exposed during the Last Glacial Maximum (LGM) (Fig. 1c)[25]. The preliminary static volume of OFG stored in the whole continental shelves of the South China Sea was roughly estimated to be $5.0 \times 10^4$ km$^3$[26], which is equal to the total annual discharge of the 100 largest rivers in the world (Fig. 1b). This could be overestimated as the calculation is made by simply summing up OFG stored in each generalized coastal segment with a 2D numerical modelling crossing entire shelves. However, their finding suggests that the South China Sea to be a hotspot area of OFG as five subaqueous deltas developed in the LDEs and adjacent shelves as shown in Fig. 1c[27]. Among them, the Pearl River is the second largest river in China in terms of water discharge[18] and the Pearl River Estuary (PRE) is located at the northern margin of the South China Sea. The runoff of Pearl River mainly drains from three tributaries, namely West River, East River, and North River (Fig. 1d) with an annual discharge of $3.43 \times 10^2$ km$^3$ y$^{-1}$ and a sediment loading of $8.5 \times 10^7$ tons y$^{-1}$ into the estuary and adjacent shelf. The total area of the Pearl River subaerial delta is about $8.7 \times 10^3$ km$^2$ with an average elevation of $3$ m[28]. These settings allow the estuaries and adjacent shelves of the Pearl River to be the natural recorder of the formation and mechanisms of the OFG.

To address the key scientific issues raised in OFG stored in the LDEs and adjacent continental shelves, we have conducted decadal offshore hydrogeology, marine seismic profiles, porewater geochemistry, and paleo-hydrogeological models in the paleo-delta and adjacent continental shelf of the Pearl River. As such, a paucity dataset of 31

offshore boreholes, associated with the high-resolution porewater geochemistry profiles, have been obtained in this work (Data and methods subsection: "onshore sampling" and "offshore porewater extraction and analysis"; Text S1 in supplementary). Paleo-hydrogeological models based on borehole logs and dense marine seismic profiles are also established to study the formation of the OFG (Data and methods subsection: "Integrated marine geophysical profiles"; "Paleo-hydrogeological modelling"; and Text S3–S5 in supplementary). We intend to answer the following fundamental questions related to OFG in the LDEs and their adjacent continental shelves: (1) Are estuaries and adjacent continental shelves of the larger river to be a hotspot to store OFG? (2) What are the provinces and formation mechanisms of OFG in the typical LDEs and adjacent continental shelves; (3) What are the primary water sources of OFG in the LDEs and adjacent continental shelves; (4) By considering the water quality indices and static volumes of OFG, can these readily accessible OFG be a potential alternative source for domestic and industrial usages, i.e., freshwater supply and raw water for desalination? This work represents the first systematic study of the OFG in a large-river deltaic estuary and its adjacent continental shelf, and the findings may have significant implications for understanding the distribution of OFG in other large-river deltaic estuaries in the world.

## Results and discussion
### The salinity of porewater in offshore boreholes
The boreholes are mainly located in the subaqueous delta of the Pearl River on the continental shelf within 50 m of the water depth (Fig. 2a), which is a suitable region for OFG development[29]. The porewater in sediment cores of offshore boreholes was extracted with Rhizon samplers, except that in HK4-10 which was extracted using a

mechanical squeezer[12,30]. Overall, the salinity of porewater in all offshore boreholes consistently decreases with depth in the PRE and adjacent continental shelf (Fig. 2b). For boreholes HK10, BH5, ZK8, and ZK5 in particular, the values of practical salinity decrease to 1.0 (the salinity of adjacent bottom seawater has an average value of 32.5) at a depth of 20 m below seafloor (m.b.s.f.), which is the upper acceptable limit as drinking water as defined by the World Health Organization. Furthermore, the salinity of porewater in sand and gravel layers for most offshore boreholes is less than 10.0, an economic threshold of OFG for the purpose of desalination[4]. However, there are two sites (PR3 and HK9-10) near Hong Kong with porewater salinity greater than 15.0. This phenomenon may be caused by geologic heterogeneity or anthropogenic activities. This offshore area has been impacted by intensive offshore engineering projects, including cross-sea bridges and tunnels, navigation channels, sea sand dredging activities, and large-scale nearshore land reclamation projects[6,11], all of which may create some saltwater infiltration into the offshore aquifers. Fortunately, this relatively high salinity OFG seems local and isolated since other boreholes (EH1 and HK10) around these two holes have salinity much lower, for example, EH1 has a salinity as lower as 6.2 and the basal aquifer of HK10 has an average salinity of 5.0.

The salinity of porewater at the top of the offshore boreholes (near the bottom seawater) is influenced by the overlying seawater and varies from 30.0 to 35.0, depending on the offshore distance. The values of salinity lower than the standard seawater is caused by the dilution effects from the freshened plume of the Pearl River discharge[31]. However, most offshore boreholes are within 55 km of the shoreline and show that the OFG is widely developed in the Pearl River Estuary and adjacent continental shelf. For example, the salinity of porewater in offshore borehole ZK3 still remains at 8.0–9.0 down to 27.5 m.b.s.f., and the aquifers in the outer shelf is still connected with the nearshore aquifers. This suggests the OFG may extend further to the southeast. Furthermore, most offshore boreholes drilled for hydrogeochemical profiles for this study (Fig. 2) do not penetrate the entire Quaternary formation and the salinity still decreases as the depth increases, so the bottom boundary of OFG should be deeper than that exposed by the boreholes in Fig. 2.

## Subaqueous delta and paleochannel systems

According to the offshore boreholes ZK1, ZK3, and BH1-13, the occurrence of OFG in the PRE and adjacent continental shelf are closely related to the offshore Quaternary aquifer systems and buried paleochannels[4,22]. Therefore, the delineation and interpretation of sedimentary and seismic facies, together with Quaternary geochronological data and buried paleochannel morphometric parameters, can provide the foundation for understanding the emplacement mechanisms and potential distribution associated with OFG in this region. The offshore Quaternary strata in the seismic reflections are usually delineated by continuous, high-amplitude, and mid-strong reflections and labelled T20, which can be continuously tracked in the study area[32–35] (see Fig. S4, supplementary materials). The reconstructed isopach map of Quaternary strata of the Pearl River subaqueous delta is shown in Fig. 2a based on the previous studies[28,33,36,37]. The thickness of Quaternary deposits gradually increases from the estuary to the adjacent shelf (to ~20.2° N), with a maximum thickness of 400 m.b.s.f. at a water depth of 120–150 m. The area of the subaqueous delta is about $3 \times 10^4$ km² with a regular fan shape. The LGM occurred ~20,000 years ago[38,39] when the mean sea level in this area was about 120 m lower than that at present conditions (Fig. 1a), so most areas of the subaqueous delta on the continental shelf were once exposed to the land surface with river networks, lakes, and rainfall infiltration.

The spatial distribution of buried paleochannels within the study area was also reconstructed based on the high-resolution marine seismic profiles and offshore boreholes logs. In summary, the buried paleochannels show a typical characteristic of discontinuities and "V"

or "U" shape downcutting in the seismic facies[40] (see Fig. S4a, supplementary materials). According to the interpreted results shown in Fig. 2a, the paleochannels in the subaqueous delta are dominated by a NS or NW trend with a maximum length of ~205 km on the continental shelf. The infills of the fluvial sediments in the buried paleochannels are mainly dominated by gravel and coarse sand (Fig. S2 and Fig. S4a, supplementary materials) but are usually covered with much more low-permeability materials, i.e., clay or silt which can be regarded as a complete preferential flow channel or offshore sub-seafloor confined aquifer[22].

## Estimation of potential static volume of OFG

The Pearl River subaqueous delta can be divided into inner shelf and outer shelf and the boundary between them is roughly along the water depth of 40 m[41]. The static volume of OFG in the inner shelf is estimated first where abundant boreholes and porewater profiles are available. From the perspective of water resource exploitation, only the OFG bodies stored in sand and occasional gravel layers (hereafter referred to as sand layers or aquifers) is incorporated into the calculations, because the porewater in the clay or silt cannot be easily extracted by pumping wells in the field[12]. The three-dimensional (3D) morphological distribution and volume of the aquifers in the inner shelf is cross-validated by high-resolution marine sub-bottom and single-channel seismic profiles together with abundant offshore boreholes (Fig. 3a). These data are processed in the Groundwater Modelling System software (GMS 10.5.6) with geostatistical tools. Furthermore, the distribution of sediment basement depth is also recognized by multi-channel seismic profiles (Fig. 3b). Two widely distributed offshore aquifers above the basement are identified. The shallower aquifer is dominated by fine sand, and the deep aquifer is dominated by medium to coarse sand (Figs. S6a, b, supplementary materials). According to the results of particle size analysis of the core samples and previous studies[6,12] (Fig. S7, supplementary materials), the average porosity of the fine sand and medium to coarse sand are set to 0.35 and 0.30, respectively. Therefore, the static volume $V_{s1}$ of OFG stored in the pores of sand layers in the inner shelf is calculated to be 61.7 km³, and the 3D morphological distribution of the OFG for this region is shown in Fig. 3c.

We then further estimated the potential static volume of OFG in the whole subaqueous delta. The OFG must exist beyond the inner shelf because the salinity of porewater in boreholes ZK3, ZK4, and BH1-13 in the margin of the inner shelf is only at 1.0–9.0, and the offshore aquifers revealed by boreholes in the subaqueous delta on the outer shelf are connected with the nearshore aquifers (Fig. S6a, supplementary materials). A set of 2D hydrogeological models based on the cross-section A-B are built to estimate the volume of OFG in the whole subaqueous delta (Data and methods: "Paleo-hydrogeological modelling"), with similar model settings to other studies when calculating OFG volumes[7,29,42,43]. Furthermore, the Pearl River subaqueous delta is generalized into a fan with a radius of 250 km and a central angle of 97° to simplify the calculation of OFG volume with a polar coordinates integration (Fig. S6c, supplementary materials)[44]. Comprehensive strata log diagram and geological boreholes in the Pearl River subaqueous delta (Fig. 2) also indicate that the stratigraphic structure in the outer shelf is similar to the cross-section A-B[32,33,45].

Our hydrogeological model results suggest that the majority of the OFG is emplaced during the LGM (Fig. 4a). The maximum offshore extent for a present-day OFG in the Pearl River subaqueous delta, as estimated by hydrogeological models, is about 250.6 km (Table 1). The total volume of OFG estimated by the base case (Fig. 4b) is 586.2 km³ with an offshore distance of 221.8 km. When the hydraulic conductivities of the clay layers in the hydrogeological model are assumed to be $10^{-10}$ to $10^{-8}$ m s$^{-1}$, the OFG volume changes from 597.4 to 472.5 km³ (Fig. 4c, d). This result shows that when the clay layers, especially the top layers are less permeable, the offshore aquifers tend

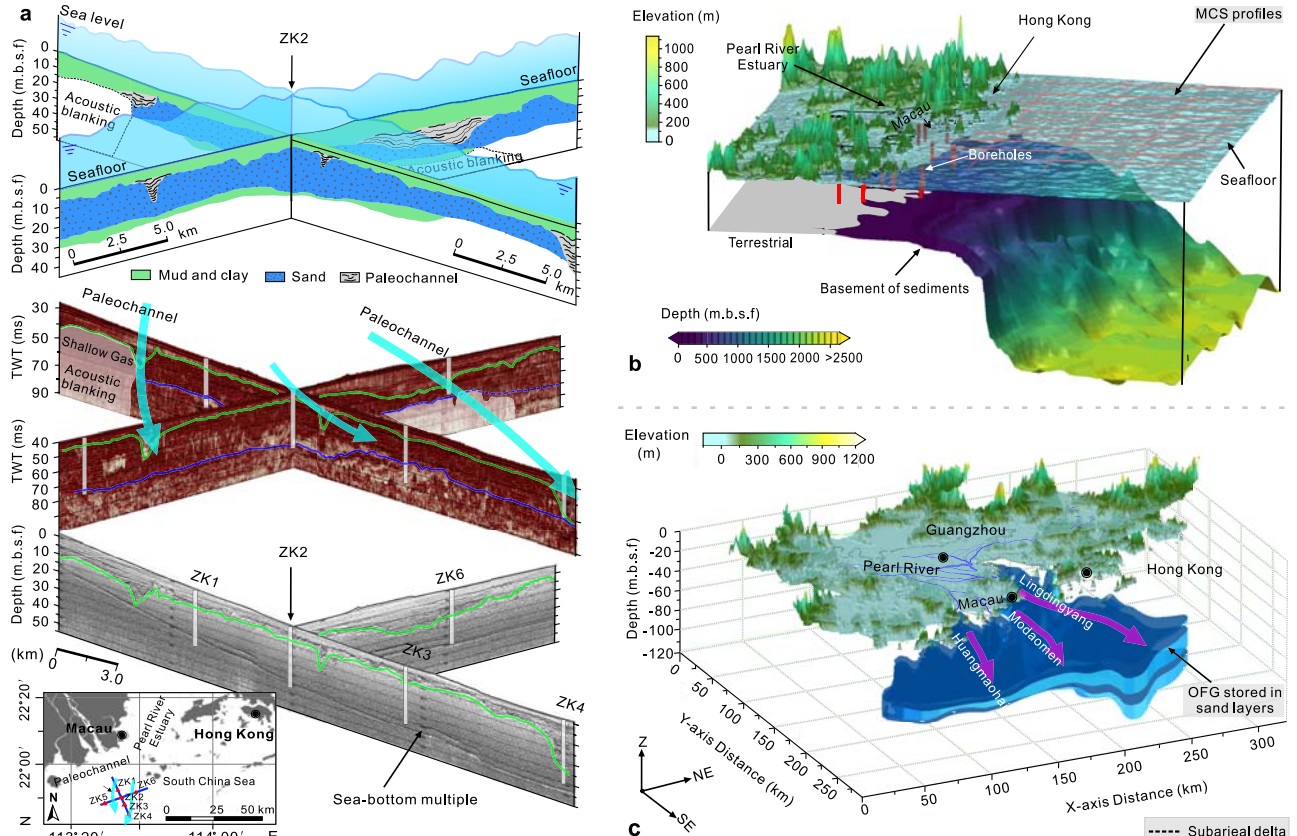

**Fig. 3 | Integrated offshore seismic-core interpretation fence diagram and three-dimensional (3D) geological models for the Pearl River estuary (PRE) and adjacent continental shelf. a** The interpreted offshore stratigraphic units and buried paleochannels using a cross-validated method that integrates the single-channel seismic profiles (middle subgraph) and sub-bottom seismic profiles (bottom subgraph) with geological boreholes (TWT, two-way travel time). **b** Spatial distribution map of basement as interpreted from the multi-channel seismic (MCS) profiles (red lines) (more information can be found in Fig. S5, supplementary materials). **c** The interpreted 3D distribution of offshore freshened groundwater (OFG) stored in sand layers for the nearshore region. The topography and offshore bathymetry data in Figs. 3b, c are sourced from GEBCO (https://download.gebco.net).

to be highly resistant to the intrusion of overlying seawater; hence, more OFG can be sequestered in the offshore aquifers (Fig. 4c). The effect of hydraulic conductivity of the shallow fine sand layers on the development of the OFG volume is similar to the clay layers. As shown in Fig. 4e, f, when the hydraulic conductivity of the fine sand is set to $1 \times 10^{-5}$ m s$^{-1}$, the calculated volume is 605.2 km$^3$, but decreases to 572.5 km$^3$ as the hydraulic conductivity increases to $1 \times 10^{-4}$ m s$^{-1}$ for the fine sand. The deeper medium to coarse sand layers are basically separated from the overlying seawater due to the low permeability of the top clay or silt layers, and all scenarios show a persistent increase in the volume of OFG when the hydraulic conductivity of the basal aquifer is increased (Fig. 4g, h). These comprehensive scenario analysis by the hydrogeological model leads to an average OFG volume estimate of $575.6 \pm 44.9$ km$^3$.

To compare the static OFG volume in the PRE and adjacent continental shelf with other passive continental margins, the average volume of OFG per km ($V_{FT}$) calculated by the hydrogeological model is equivalent to $4.7 \pm 0.44$ km$^3$ km$^{-1}$, while the volume of OFG in other passive continental margins mainly ranges between 1.0 and 4.8 km$^3$ km$^{-1}$, i.e., 3.24–4.78 km$^3$ km$^{-1}$ offshore of Canterbury[7], 1.7 km$^3$ km$^{-1}$ offshore of New England[42], 4.4 km$^3$ km$^{-1}$ offshore of New Jersey, 1.0 km$^3$ km$^{-1}$ offshore of Jakarta, and 3.1 km$^3$ km$^{-1}$ offshore of Gippsland[4].

## Origin of OFG in the LDE and adjacent shelf

The values of $\delta^{18}$O and $\delta^2$H together with the Cl of the porewater can also be used as the indirect constraints for the origin of OFG[3,14]. As shown in Fig. 4i, j, the $\delta^{18}$O and $\delta^2$H values of the freshened porewater in the PRE and adjacent shelf are similar to the onshore groundwater and river water and intersect with the LMWL, suggesting a predominantly meteoric water source. The freshened porewater does not originate from the dehydration of clay minerals (a low-salinity anomaly in Fig. S3a is discussed in the supplementary materials). This is because freshening due to clay mineral dehydration leads to highly enriched $\delta^{18}$O values and highly depleted $\delta^2$H values[46]. For example, ref. [47] provide endmembers of +10‰ ($\delta^{18}$O) and −32‰ ($\delta^2$H) for the freshened porewater caused by clay mineral dehydration from the Eastern Mediterranean.

Furthermore, the unconsolidated seabed sediments in the northern margin of the South China Sea are also rich in gas hydrates, and freshened water can be released by the decomposition of buried gas hydrates (i.e., a low-salinity anomaly in Fig. S3b is discussed in the supplementary materials)[48], but this process will cause a shift to more positive values for both $\delta^{18}$O and $\delta^2$H (Fig. 4i, j), e.g., +2.5‰ ($\delta^{18}$O) and +22‰ ($\delta^2$H) at the Hydrate Ridge, Cascadia margin[49] and +1.6‰ ($\delta^{18}$O) and +8‰ ($\delta^2$H) in the eastern Pearl River Mouth Basin, northern margin of the South China Sea[50]. Therefore, Fig. 4i and j clearly show that the freshened porewater collected from the offshore boreholes in the PRE and adjacent continental shelf most likely originate from the meteoric recharge and not the decomposition of buried gas hydrates.

The volume of porewater collected from the marine sediments from offshore boreholes (usually 20–50 mL) does not reach the detection quantity limit for dating absolute ages (i.e., $^{14}$C, $^{36}$Cl, $^4$He, $^{39}$Ar, $^{81}$Kr)[3]. Therefore, whether the OFG originated from modern, or

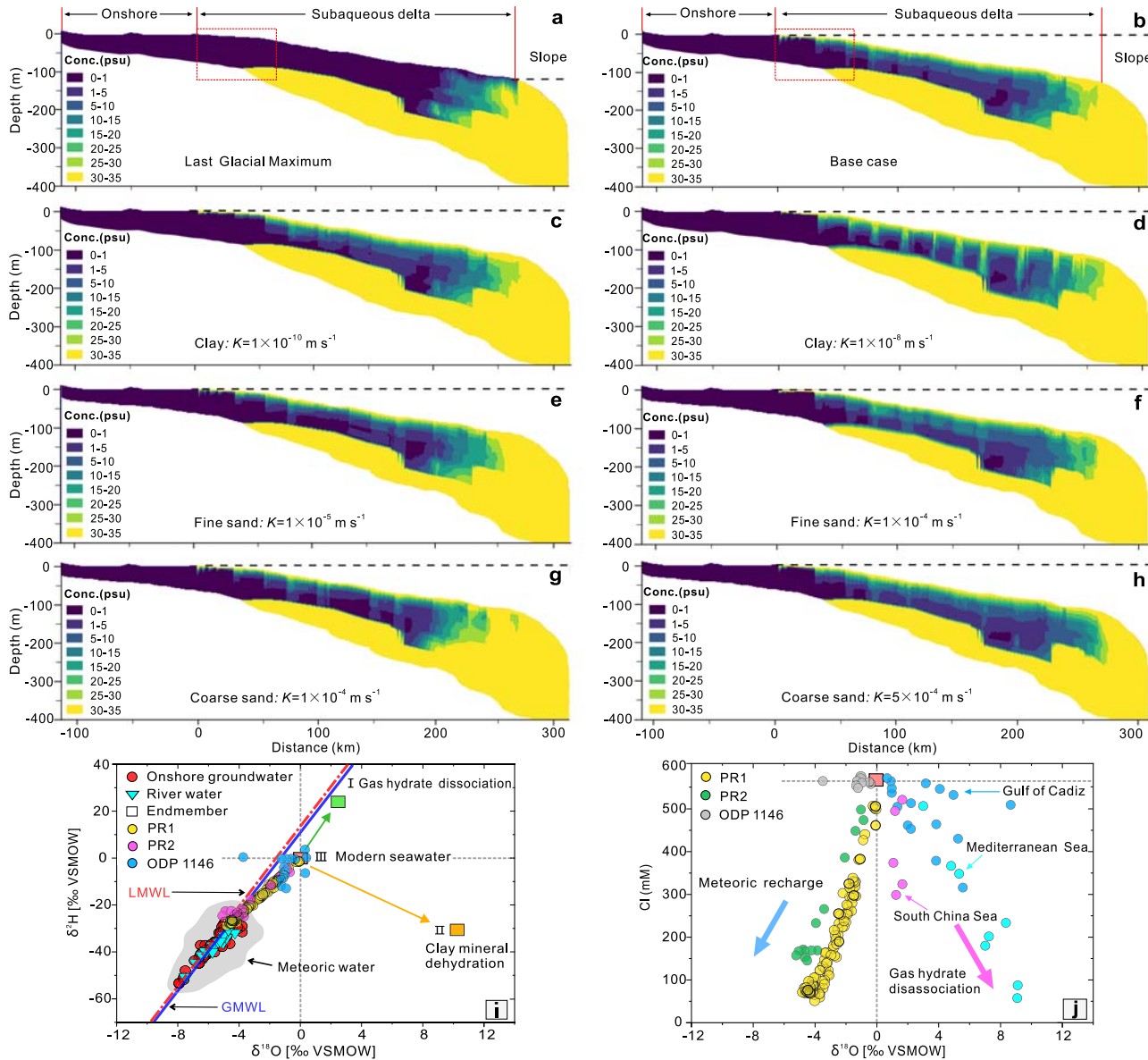

**Fig. 4 | Hydrogeological model results with different scenarios for the cross-section A-B and stable isotopic compositions of porewater with different endmembers.** The location of this cross-section is shown in Fig. 2a. **a** Computed salinity distribution at the Last Glacial Maximum. **b** Computed present-day salinity distributions for the base case. The red rectangles in **a**, **b** show the inner shelf area with an offshore distance 55 km. The dashed black line denotes the relative sea level. **c**–**h** Computed present-day salinity distributions with different hydraulic conductivities of the key hydrogeological units. **i**, **j** Plots of $\delta^{18}O$ versus $\delta^2H$ and chlorinity (Cl) for the porewater from offshore boreholes PR1, PR2, and ODP1146. The solid blue line labelled by GMWL is the global meteoric water line, while LMWL is the local meteoric water line.

paleo meteoric water is not clear only based on the stable isotopic compositions. However, considering the present-day distribution of hydraulic gradients in the Pearl River delta and adjacent continental shelf (the average elevation of the Pearl River delta plain is around 3 m), the low-lying topography may not drive the onshore precipitation recharge to flow very far seaward under the present-day conditions. The most compelling explanation for the vast OFG bodies is that the OFG was originated from the local meteoric precipitation when the offshore areas were exposed to the atmosphere at the low-stand periods since the late Pleistocene. During sea-level low-stands, topographically driven flow takes place across much of the continental shelf, where the hydraulic gradients are about an order of magnitude higher than that during high-stands. As the sea level rose rapidly, this entrapped low-salinity paleo-groundwater in the offshore aquifers had a sluggish response to the sea level rise due to the low permeability of the overlying clay and silt layers.

## Water quality of OFG in the PRE and adjacent shelf

Water quality is an important criterion to determine whether OFG can be used as potential potable water or as a raw water source for desalination. However, studies on the geochemical characteristics of direct offshore porewater samples are rare because previous studies on OFG were mainly based on indirect methods, i.e., marine geophysical profiling and numerical modelling[3,13,14]. Table S1 in the supplementary materials compares the typical water quality indices (i.e., major ions, nutrients, heavy metals, and trace elements) in the OFG with surrounding seawater and drinking water in the PRE and adjacent continental shelf. Some indices such as pH and salinity in some offshore boreholes are very close to the drinking water standards, but most exceed the limits, so directly using the OFG as drinking water is obviously not possible. However, the OFG has a very low concentration of major dissolved cations and anions (i.e., $Na^+$, $K^+$, $Ca^{2+}$, $Mg^{2+}$, $Cl^-$, $SO_4^{2-}$) compared to the overlying seawater, and so could be used as

**Table 1 | The volume of offshore freshened groundwater (OFG) in the Pearl River estuary and adjacent continental shelf estimated by hydrogeological models**

| Scenarios | Base case | Case c | Case d | Case e | Case f | Case g | Case h |
|---|---|---|---|---|---|---|---|
| $L$ (km) | 221.8 | 221.9 | 219.8 | 210.2 | 225.0 | 200.1 | 250.6 |
| $V_{FT}$ (km$^3$ km$^{-1}$) | 4.86 | 4.92 | 3.92 | 4.66 | 4.75 | 4.35 | 5.45 |
| $V_{FS}$ (km$^3$ km$^{-1}$) | 2.50 | 2.57 | 1.95 | 2.38 | 2.19 | 2.39 | 2.58 |
| $V_{TS}$ (km$^3$) | 586.2 | 597.4 | 472.5 | 605.2 | 572.5 | 575.5 | 620.3 |

$L$ offshore distance (km), $V_{FT}$ static volume of OFG per unit width in all materials (km$^3$ km$^{-1}$), $V_{FS}$ static volume of OFG per unit width in sand layers (km$^3$ km$^{-1}$), $V_{TS}$ total static volume of OFG in sand layers in the whole Pearl River subaqueous delta (km$^3$).

raw source water for desalination with associated savings in costs and energy compared to the sources of seawater with a higher TDS concentration (usually range between 31.0–33.0 in the study area).

According to the summary in Table S1, OFG has a higher $NH_4^+$ but lower $SO_4^{2-}$ and $NO_x^-$ compared to the surrounding seawater, which is due to the microbial decomposition of organic matter in the seabed sedimentary environment. For example, the organic detritus in the marine sediments especially in the buried paleochannels supports an elevated microbial metabolism. An ideal environment for microbial $NO_3^-$ reduction is thus created where $NO_3^-$ can be in ample supply to substitute for $O_2$ during organic matter degradation, eventually leading to $NH_4^+$ productions via ammonification[51]. This emplacement mechanism is similar to that responsible for the abnormally high $NH_4^+$ of groundwater in the Pearl River Delta[52]. As one of the main oxidants in marine sediments, $SO_4^{2-}$ is reduced from the oxidation of buried organic materials, supplemented by the anaerobic oxidation of methane at the sulfate-methane transition zone[53].

Heavy metals and trace elements are also essential indicators for assessing the quality of water. Recently, heavy metal pollution in estuaries and offshore seawater due to intensive anthropogenic activities has received increasing attention worldwide[54]. However, OFG is not at risk of heavy metal pollution as OFGs are mostly paleo-terrestrial groundwater and the presence of aquitards or marine units serves as the membrane to filter heavy metal from overlying seawater. The key parameters pointing to portable water quality i.e., Ba, Cd, Cr, Cu, Pb, Sr, and Zn in the OFG of the PRE and adjacent shelf are all below the drinking water standard. Remarkably, the concentration of dissolved Fe in the OFG is much higher than in the overlying seawater (i.e., the concentration of dissolved Fe is 6527.8 and 1902.1 ug L$^{-1}$ at the PR2 and PR1 sites respectively, which is much higher than the average value of 68.3 ug L$^{-1}$ in the surrounding seawater).

In summary, the systematic analysis of the geochemical characteristics of the OFG shows that such offshore freshened water bodies may not be directly used as drinking water. However, it can be used as a cost-effective raw source water for desalinization or as drinking water after minor treatment on some specific ions in the coastal cities. In particular, it can also be used as a water resource for agricultural, domestic, and industrial purposes in some coastal regions and island nations that experience severe water shortages as a result of changing climate, intensified pollution, and over-exploitation caused by population growth and urbanization.

**Implications for other LDEs and adjacent continental shelves**

OFG is believed to exist widely in LDEs and adjacent continental shelves worldwide and be characterized by a much lower salinity than seawater (Fig. 1a and Table S3 in supplementary materials). To the best of our knowledge, this is the first study to report on such a large number of offshore boreholes drilled to obtain direct geochemical evidence of OFG in an LDE and adjacent shelf. During the Quaternary period, most LDEs experienced multiple transgressions and regressions, and the subaqueous deltas are widely developed in the adjacent continental shelves;[17] the general evolutionary process is shown in

Fig. 5a–c. Sea level significantly decreases during glacial periods, and the original submerged continental shelf becomes a terrestrial environment with sedimentation dominated by coarse-grained fluvial deposits. The river networks extend further to the sea and are infilled by gravel and coarse sand (Fig. 5a, d). However, during transgressions, fine-grained marine sediments, dominated by clay, silt, or fine sand are deposited, and the original subaerial delta will be submerged and transformed into a subaqueous delta (Fig. 5b, e). Generally, this depositional cycle will repeat many times due to the sea level fluctuations during the Quaternary period (Fig. 5c, f). The result is the formation of an offshore interlayered aquifer-aquitard system in this paleo-geological sedimentary environment.

The findings of this study provide insights into understanding the evolution and occurrence of OFG in the PRE and adjacent continental shelf. Some favourable geological factors in LDEs are required to form OFG and can also indicate priority areas for detecting OFG at the preliminary stage. Specifically, these include (1) an extensive sedimentary system dominated by sand and gravel to provide a large area to store considerable amounts of fresh groundwater; (2) confining and semi-confining formations dominated by clay and silt so that vertical mixing of groundwater in different aquifers does not occur or is slow; (3) large buried paleochannel systems with relict fluvial channels that are infilled with high-permeability sediments and act as preferential pathways to provide a hydraulic connection between the onshore and offshore aquifers; and (4) an adequate hydraulic gradient of the onshore freshwater source, which can force the fresh groundwater discharge to the sea against seawater intrusion.

## Methods

From May 2002 to November 2021, we collected data from 31 offshore boreholes in the PRE and adjacent shelf of the northern South China Sea. All offshore boreholes were drilled by the ocean scientific drilling vessel "*Haiyang Dizhi*-10", except boreholes HK4-10 which were drilled in earlier projects in the east of the PRE and analyzed in our previous studies[11,12,30]. Considering the scarcity of offshore boreholes, different types of marine seismic profiles were also collected to provide indirect constraints on the lithology and permeability of the seabed sediments, distribution of the buried paleochannel systems, and geometry of the depositional units and faults. The locations of the onshore sampling sites, offshore boreholes, and marine seismic profiles can be found in Fig. 1c. Below, we describe the onshore groundwater and river water sampling, offshore porewater extraction, data analyses, and interpretation of the integrated marine geophysical profiles in the PRE and adjacent continental shelf.

### Onshore sampling

Systematic onshore groundwater sampling was continuously conducted from 2006 to 2022 in the Pearl River Delta (PRD)[18]. Groundwater samples from boreholes, piezometers, and farm wells as well as porewater samples from the sediment cores of boreholes were collected. As part of the comprehensive sampling, river water in the PRD was also individually sampled in January 2022. All water samples for chemical analysis were filtered with 0.45 μm syringe filters and analyzed in situ for salinity and pH with portable probes

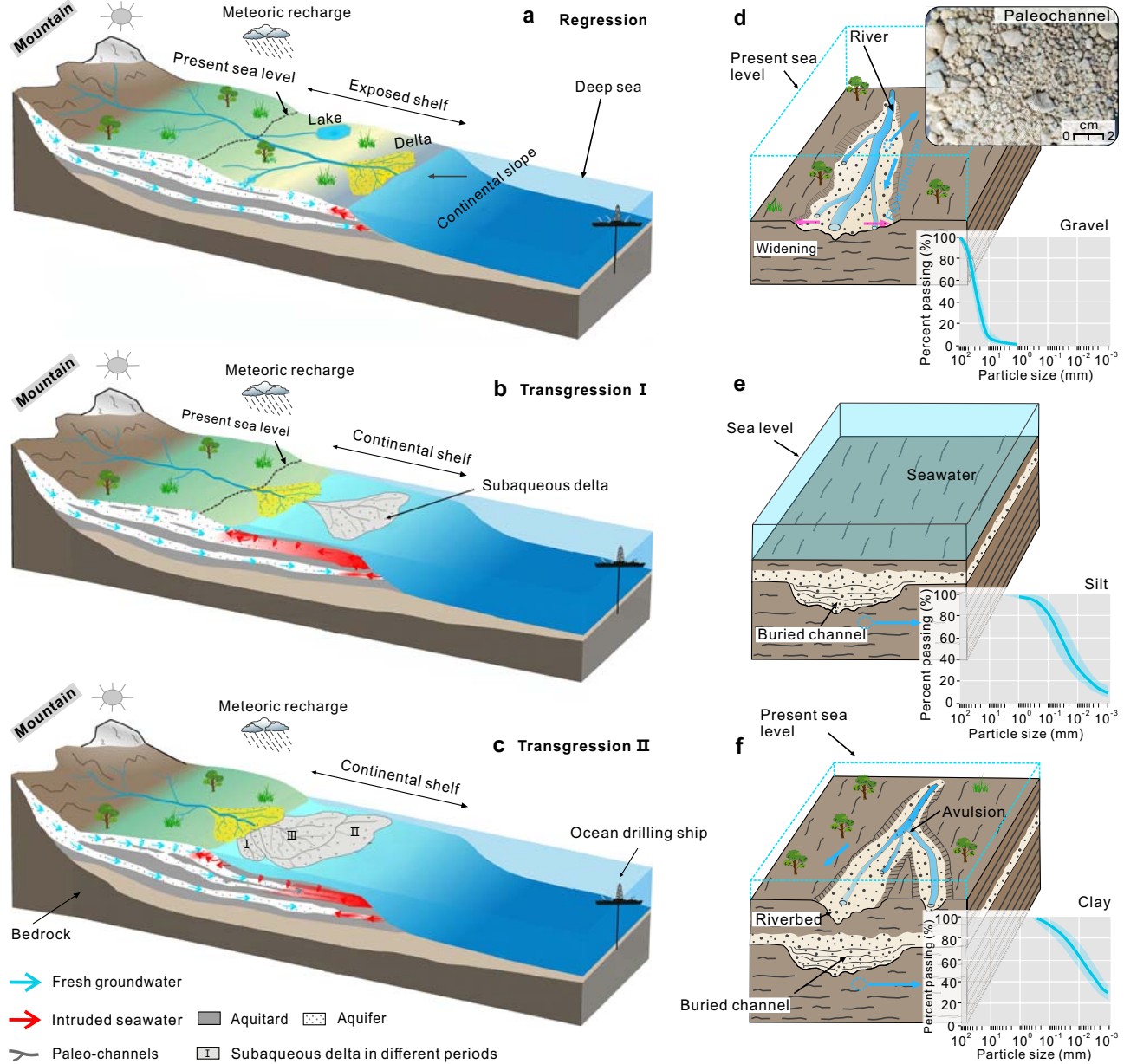

**Fig. 5 | Schematic illustration of the geologic stratigraphy associated with delta progradation and aggradation, and the key groundwater flow in a large-river delta-front estuary (LDE) and adjacent continental shelf. a** Lower sea-level during glacial periods promote the further penetration and recharge of groundwater below the continental shelf under a higher hydraulic gradient, whereas incised rivers and lakes provide a driving force for topography-induced flow systems. **b** As the sea level rises, the shoreline moves inland, and the subaerial delta becomes a subaqueous delta. The intruded seawater (red arrows and shadows) migrates landward as well as downward, while the flow of fresh groundwater (blue arrows) stagnates. **c** Due to sea level fluctuations during Quaternary, such depositional cycles repeat many times in the LDE and adjacent continental shelf. **d**–**f** Conceptual models proposed to explain the formation of buried paleochannels during transgressive-regressive cycles; the inset shows the deposits photography in buried paleochannels and particle size distribution curves for different materials.

(Hanna Instruments (Pty) Ltd), and then stored at 4 °C until further analysis. A 2% $HNO_3$ solution was added in the field to cation ($Na^+$, $K^+$, $Ca^{2+}$, and $Mg^{2+}$) samples to prevent precipitation. The analysis of water samples included conventional hydrochemical parameters (i.e., salinity, pH, cations, and anions) and stable isotopes ($\delta^2H$ and $\delta^{18}O$). Major cations and anions were determined by ion chromatography (Thermo Scientific Dionex ICS-1100) in the Hydrogeology Laboratory of the University of Hong Kong with analytical errors of less than 3%. Values of $\delta^2H$ and $\delta^{18}O$ were measured with off-axis integrated cavity output spectroscopy (OA-ICOS) and a Triple Isotope Water Analyzer (TIWA-45EP) at the State Key Laboratory of Marine Geology, Tongji University. The standard deviations of all water samples and standards were less than 0.5 and 0.1‰ for $\delta^2H$ and $\delta^{18}O$, respectively.

## Offshore porewater extraction and analysis

Porewater was extracted from the cores of the offshore boreholes at 20 cm intervals using Rhizon samplers in the laboratory of the ocean drillship as much as possible. Generally, a Rhizon sampler consists of four parts: (1) a thin tube comprised of a hydrophilic membrane, (2) an unbending wire to support the tube, (3) a flexible hose to pass water from the tube, and (4) a connector connecting the subsequent collection syringe[55]. The thin and porous tube is inserted directly into an intact sediment core, and a 20 mL syringe is attached to the connector. The vacuum in the syringe is the main driving force for the extraction of the porewater from the sediment core. Porewater then passes from the sediment through the porous tube and flexible hose into the collection syringe. Furthermore, a three-way valve is added between the

connector and syringe to facilitate multiple samplings in one position (see schematic diagram in Fig. S1, supplementary materials). Given a sufficiently small tube pore size (0.15 μm), the Rhizon sampler also serves as a filter, removing microbial and colloidal contamination. The heavy metals and trace elements of the offshore porewater samples were measured using inductively coupled plasma mass spectrometry (ICP-MS) (Agilent 7900 Series, USA) at the Joint Laboratory for Chemical Geodynamics of the University of Hong Kong with indium as an internal standard. All samples were analyzed in triplicate, and the relative standard deviation for analytical precision was less than 5%. The other pre-treatments and hydrochemical analyse of the porewater were similar to the onshore samplings as described earlier.

### Integrated marine geophysical profiles

Marine seismic profiles, mainly from four separate cruises, were used to provide indirect constraints on the distribution of paleochannels, faults, and formations (Fig. 1c). The high-resolution sub-bottom seismic (SBS) profiles with a total length of ~1000 km inside the PRE were obtained in 2001 by the South China Sea Institute of Oceanology, China Academy of Sciences, using *GeoPulse* and *GeoChirp* by *Geoacoustic Corp* with a shot interval of 500 ms and a recording scale of 1000 ms. Other 1000 km single-channel seismic SCS profiles were obtained by the Guangzhou Marine Geology Survey and China Geological Survey in 2004 and 2005, using an *Elics Corp*. system with a 250 ms recording interval[40]. Furthermore, to characterize the geometrics of faults and basements in the PRE and adjacent continental shelf, 13 MCS (24-channel) profiles were collected using a MicroEel Analog Seismic Solid Streamer with a record length for each shot of 6 s and 1 ms sampling rate[56]. As part of the research program of the Southern Marine Science and Engineering Guangdong Laboratory (Zhuhai), 19 high-resolution SBS profiles (~1600 km) were also obtained in September 2020 from the PRE and adjacent continental shelf, using an *Edgetech-512i* towfish and *Edgetech-4200p* chip sub-bottom profiler system. The vertical and horizontal resolutions were 0.2 and 0.5–1.0 m, respectively.

Different types of marine seismic profiles play different roles considering their signal characteristics; for example, the SBS signal has a higher frequency than the SCS signal and can easily penetrate the mud and clay layers but will quickly dampen in sand and gravel materials. However, the SCS signal can penetrate the clay and sand layers but the signal interpretation is difficult and suffers from non-uniqueness[57]. Therefore, cross-validation between different marine seismic profiles is necessary. The objectives of the high-resolution SBS and SCS here are mainly two-fold: (1) to depict the geometrical morphometrics and spatial distributions of buried paleochannels and (2) to constrain and trace stratigraphic structure, especially the sand and silt or clay layers (Fig. S4, supplementary materials). Furthermore, because the penetration depth of the MCS signal is much greater than for the other two types[56], the MCS profiles in this study can be used to evaluate the distribution of sediment basements and potential faults (Fig. S5, supplementary materials). All marine seismic profiles were interpreted by benchmarking the log information of the offshore boreholes. The positions of different types of marine seismic profiles can be found in Fig. 1c.

### Paleo-hydrogeological modelling

To model the distribution of groundwater salinity in the PRE and adjacent continental shelf, we used the variable-density groundwater flow and coupled salt transport modelling code SEAWAT[58] to set up a 2D model for the described transect (A-B in Fig. 2, more information can be found in Fig. S6 in supplementary materials). It is highly probable that large OFG was formed at low sea-level tens of thousands of years ago when the continental shelf was exposed to the atmosphere[3,4,42]. For such reasons above, the timescale considered in this study stretches beyond one full glacial-interglacial cycle to determine not only the current situation but also the temporal dynamics of the regional groundwater systems similar to some previous studies[21,43]. Therefore, the model considered sea level variations of 120 m over a 125 kyr period. The model domain is divided into 840 columns with 100 m wide cells in the horizontal, and up to 205 layers with an average thickness of 2 m. The bottom and left boundaries of the model domain are set to no-flow boundaries. In the offshore domain, the uppermost model cells and the rightmost column are assigned a specified head boundary, which equals the sea level elevation according to the reconstructed eustatic sea-level curve, with a concentration of 35 g L$^{-1}$. For nodes in the land above sea level, we imposed a specified flux at the uppermost cells to simulate the rainfall recharge. As no long-term precipitation record exists for the Pearl River delta in most of the period of the past 125 kyr, we choose to apply a constant uniform recharge of 1.25 mm d$^{-1}$, which is equal to about 25% of the current long-term precipitation average[52].

We approximate the continuous sea level changes by 31 subsequent stress periods[59]. These stress periods are chosen to capture the fluctuations of sea-level values and stretch over fixed time periods. The parameters and boundary conditions in each stress period are fixed. Since sea-level drop occurs in a large part of the glacial-interglacial period and the dropping rate is much lower than that of the sea-level rise period. The stress periods of the sea-level drop are longer than that during the sea-level rise (5 and 2 kyr, respectively). These treatments ensured that the effects of sea-level fluctuation on the groundwater flow and salinity dynamics are captured with enough details as suggested by previous studies[43]. The models are initially set to run for the time duration of the full glacial-interglacial cycle (125 kyr) with modern fixed sea-level conditions. This approach is adopted to estimate the starting groundwater salinity conditions before simulating the full glacial-interglacial cycle with the fluctuating sea-level boundaries.

The hydraulic properties used in the hydrogeological models are based mainly on the recent research for the PRE and adjacent continental shelf[12,37,60] and other studies in similar deltaic systems[3,21,61,62]. There are four hydrogeological units used in the cross-section: clay, silt, fine sand, and medium to coarse sand (Fig. S6, supplementary materials). The values of horizontal conductivities of the four units for the base model vary between $1 \times 10^{-9}$ m s$^{-1}$ and $2.5 \times 10^{-4}$ m s$^{-1}$ (Table S2, supplementary materials). The vertical hydraulic conductivities of the hydrogeological units are assumed to be 1/3 of the horizontal hydraulic properties similar to previous studies[61]. Longitudinal dispersivity is set to 50 m for all units, and the horizontal and vertical transversal dispersivities are assumed to be 5 and 0.5 m, respectively. These values are consistent with the models of similar previous studies[61,63,64]. We assumed a molecular diffusion coefficient of $8.64 \times 10^{-5}$ m$^2$ d$^{-1}$, and specific storage and specific yield are equal to $10^{-6}$ m$^{-1}$ and 0.25 for all layers, respectively. In order to determine the potential range of the static volume of OFG in the PRE and adjacent continental shelf, seven sensitivity analyses of the hydraulic conductivities of the key hydrogeological units were performed (Table 1).

## Data availability

All data generated or analysed during this study are included in the published article (and its supplementary information file). The raw geochemical data for porewater from sediment cores are available from the corresponding author and first author upon reasonable request.

## Code availability

The SEAWAT version 4 code is available from USGS: https://water.usgs.gov/ogw/seawat. Free software packages Generic Mapping Tools (GMT) version 6 (www.generic-mapping-tools.org) is used for creating some figures.

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

## Acknowledgements

This study was supported by grants from the Key Program of the National Science Foundation of China and Innovation Group Project of Southern Marine Science and Engineering Guangdong Laboratory (Zhuhai) (No. 42130702, 311021004) to J.J.J., and the Hong Kong General Research Fund (No. 17307521) to J.J.J. Many thanks are given to Shengchao Yu, Hongbin Liu, Jingye Xian, Long Xi, Mei Chen, Yanqiong Huang, and Meiqing Lu for their kind help in the sampling campaigns and sample analysis. Appreciation is also given to the master, technicians, and crew of the ocean drilling vessel *Haiyang Dizhi*-10 for their support during the cruise.

## Author contributions

C.S. and J.J.J. designed the study, interpreted the results, and prepared the manuscript. C.S. carried out the fieldwork, processed and analysed the data. X.L. took part in designing the study and reviewing the manuscript. J.Z. participated in the offshore sampling and conducted analyses of some porewater samples. L.J. provided the raw data for some offshore boreholes used in the study. J.C. provided and analysed the offshore multiple-channel seismic data. All co-authors edited the manuscript and provided suggestions on how to improve the analyses.

## Competing interests

The authors declare no competing interests.
