## [Peer Review File · Nature Communications]

Offshore freshened groundwater in the Pearl River estuary and shelf as a significant water resourceEditorial Note: Panels on p12 of this file reproduced respectively from Allison, M. A., Khan S. R., Goodbred S. L., & Kuehl S. A. (2003), Stratigraphic evolution of the late Holocene Ganges–Brahmaputra lower delta plain, *Sedimentary Geology*, **155**(3), 317-342, Copyright (2002) with permission from Elsevier; and Kuang, X., Jiao J. J., & Wang Y. (2016), Chloride as tracer of solute transport in the aquifer–aquitard system in the Pearl River Delta, China, *Hydrogeology Journal*, **24**(5), 1121-1132, with permission from Springer Nature.

REVIEWER COMMENTS

Reviewer #1 (Remarks to the Author):

Major comments

This work presents pore water data on the inner shelf off the Pearl River estuary (PRE) and extrapolated to the outer shelf to estimate the volume of freshened groundwater in this region. The presence of freshened groundwater in large river deltaic estuaries has been demonstrated/suggested in a few systems, including the Pearl River Delta. This study confirmed the presence with pore water data on the inner shelf off the PRE. However, the extrapolation from the pore water data on the inner shelf to a scenario on the outer shelf where no pore water data are available is questionable. Even on the inner shelf, although in Core PR3 sands and gravels are the main components in the lower sediments, salinity is greater than 10 at these depths (Figure S3). So is the salinity at the lower sediment in Core HK9. Thus, the extrapolation that the authors made in assuming that sediments at depths where sands were present on the outer shelf have freshened groundwater and no more saline mixing is kind of speculative and needs further validation. The estimation of the volume of freshened groundwater based on such extrapolation, therefore, has large uncertainty, which needs to be assessed. Practical salinity is unitless (PSS-78) and should not be followed by PSU.

Minor comments

Line 36: “dynamics” is ambiguous in its meaning here. Not sure what “freshwater source dynamics” means here.

Line 70 : what is “the coastal zone”? Specify.

Line 109: change “located” to “is located”

Line 111: change “the three” to “three”

Line 126: change “is” to “are”

Line 128: change “is” to “are”

Line 129: change “are” to “can”, remove “can”

Line 141: change “in the continental shelf” to “on the continental shelf”

Line 142: change “South China” to “southern China”

Line 154: “a unitless quantity equivalent to g/kg” is misleading since absolute salinity is in g/kg, but practical salinity is not equivalent to absolute salinity.

Figure 2: subplots BH5 and HK10 should have x axis labels and titles; subplots PR1 and BH5 should have y axis titles.

Line 189: change “in” to “on”

Line 198: change “regard” to “regarded”

Line 214-216: how did the authors jump from the calculated total static volume of the OFG of $61.7 \times 10^9 \text{ km}^3$ to the potential static OFG volume of $523.3 \times 10^9 \text{ km}^3$? It is confusing that the value of the potential static OFG volume just appeared without any calculation or explanation. Why did the two volumes differ? What is the purpose of presenting Eq. (1) here since the total volume is not what the authors need?

Reviewer #2 (Remarks to the Author):

This is an excellent manuscript worthy of publication in Nature Communication. This is the first study to document the existence of extensive vast ($> 500 \text{ km}^3$) offshore fresh to brackish water resources at shallow depths (50m below the sediment water interface) in the South China Sea in an active delta system. It is highly likely that these freshwater to brackish water resources extend to greater depths. The authors argue that rapidly avulsing deltas will sequester vast amounts of fresh to brackish water found in sub-aqueous deltas during Pleistocene sea level low stands. I do have a number of suggested revisions that I believe would improve the manuscript:

In general, I found the attempts to portray the three-dimensional topography as not very helpful

(e.g. Fig 1b, 3b,c) given the large topographic range. The authors might consider plane-view plots instead. It would be helpful to see contours of the shallow (0-100m) bathymetry where the offshore wells are located.

Along these lines, not much attention to well water depth is given in this manuscript. It is difficult to see if there is any correlation between seawater depth and salinity as suggested by Fig. 9 in Person et al. (2017). In general, shallower seawater depths areas would have been exposed longer to meteoric inputs of recharge.

Figure 2 presents the select salinity profiles (i.e. the freshest wells). There are others that are more salty. All salinity profiles should be included. Perhaps, if necessary, in supplemental materials.

Not much attention in the manuscript is paid to river avulsion and it's potential roll in preserving offshore freshwater (schematic, Fig. 4). Active deltas have a lot of channel switching...see for example, Fig. 13 from Allison et al. (2003). Could there be a correlation between groundwater salinity and the age of a given delta front?

The article Zhang et al. (2011) is not accessible to English readers so it is hard to evaluate this citation. That is a pity.

Line 104-105 The Zamrsky calculations of offshore freshwater are almost certainly an overestimate. These authors used unrealistically optimistic representations of continental shelf hydrogeology.

In the discussion section, the authors argue the utility of plots of stable isotopes. It would be nice to see a plot of $\delta^{18}O$ verses chloride concentrations. Yet, I don't see this plotted for the 31 offshore wells presented in this study. So, I don't find Figure 4 particularly important.

The authors seem to be arguing here that active delta systems are excellent venue for the sequestration of offshore freshened groundwater. Early discoveries of offshore freshwater were found along passive margins with relatively slow rates of riverine inputs of sediments and surface water (Hathaway et al. 1979). Interestingly, much of the freshwater is sequestered in pre-Pleistocene sediments (Meisler et al. 1984) at depths of about 200m. Here, the freshwater is found in shallow, young sediments. It might be interesting to compare the volumes of freshwater stored in the Pearl River delta and the USA North Atlantic Shelf.

Are there correlations between distance from a paleo- channel and the salinity of offshore wells? It doesn't look so obvious from Fig. 2.

What were the fluid pressure conditions during drilling? That is, were there artesian conditions? There might not be due to the large number of unmapped paleo-channels.

Editorial comments:

Line 101: change "since" \diamond "during"

Line 106: change "still clues" \diamond "suggests"

Line 107: delete "enormous"

Line 109: delete "representative"

The font in Fig. 1a is really small. Suggest making Fig 1b plane view.

References

Allison, M.A., Khan, S.R., Goodbred Jr, S.L. and Kuehl, S.A., 2003. Stratigraphic evolution of the late Holocene Ganges-Brahmaputra lower delta plain. *Sedimentary Geology*, 155(3-4), pp.317-342.

Hathaway, J.C., Poag, C.W., Valentine, P.C., Manheim, F.T., Kohout, F.A., Bothner, M.H., Miller, R.E., Schultz, D.M. and Sangrey, D.A., 1979. US Geological Survey Core Drilling on the Atlantic Shelf: Geologic data were obtained at drill-core sites along the eastern US continental shelf and slope. *Science*, 206(4418), pp.515-527.

Lofi, J., Inwood, J., Proust, J.N., Monteverde, D.H., Loggia, D., Basile, C., Otsuka, H., Hayashi, T., Stadler, S., Mottl, M.J. and Fehr, A., 2013. Fresh-water and salt-water distribution in passive margin sediments: Insights from Integrated Ocean Drilling Program Expedition 313 on the New Jersey Margin. *Geosphere*, 9(4), pp.1009-1024.

Meisler, H., Leahy, P.P. and Knobel, L.L., 1984. Effect of eustatic sea-level changes on saltwater-freshwater relations in the northern Atlantic Coastal Plain (Vol. 2255). US Government Printing Office.

Person, M., Wilson, J.L., Morrow, N. and Post, V.E., 2017. Continental-shelf freshwater water resources and improved oil recovery by low-salinity waterflooding. *AAPG Bulletin*, 101(1), pp.1-18.

What would a marine electromagnetic field campaign (e.g. Gustafson et al. 2019; Attais et al. 2022) reveal in this region?

Reviewer #3 (Remarks to the Author):

the study represents an interesting study about offshore freshened groundwater in the Pearl River Estuary. The authors show various data sets, including seismic data (with different resolutions), borehole data, and hydro-geochemical data.

The case study is very interesting, but in my opinion a better description of the available data is needed.

In Section 3, the authors report an estimate of the potential static volume of OFG. In my opinion, it is not clear how many sand layers they have considered. To estimate the total volume, they assigned a porosity value to each layer. No information is provided on this. How have the authors chosen this parameter? What values have the authors adopted? From boreholes? It is important to add this data. An estimate of the error should be given.

The quaternary isopachic map was created by using seismic data because no borehole reached the base of the Quaternary sediments. I have some questions: what procedure has it been adopted to interpolate the data obtained from seismic lines? How have you translated the seismic section from time to depth? With a unique velocity profile? What kind of velocity? RMS velocity translated in terms of interval velocity? This is not clear from the text.

In addition, it is better to replace "quaternary isopachic map" with "quaternary isopach map".

In Supplementary Material TS3, a basement is reported on seismic sections. Is it possible to provide also the interpretation of the base of Quaternary sediments?

The authors should follow journal guidelines for formatting references.

The attached file contains additional comments.

Response to Comments by the Reviewers

Ref: “*Offshore freshened groundwater in large-river deltaic estuaries and adjacent shelves as a significant water resource*” (ID # NCOMMS-22-46287) by Chong Sheng, Jiu Jimmy Jiao, et al.,

(We first quote the **Comments** and then give the **Responses**, the red line number refers to the ***Revised Manuscript with changes marked**)

- **Accepted** = we **agree** with the reviewer’s comment and have modified the text accordingly.
- **Well taken** = we **partially agree** with the reviewer’s comment and have partially modified the text.
- **Not agree** = we **do not agree** with the reviewer’s comment and provide an explanation and justification of our approach.

Major comments from Reviewer #1

General Comment a): This work presents pore water data on the inner shelf off the Pearl River estuary (PRE) and extrapolated to the outer shelf to estimate the volume of freshened groundwater in this region. The presence of freshened groundwater in large river deltaic estuaries has been demonstrated/suggested in a few systems, including the Pearl River Delta. This study confirmed the presence with pore water data on the inner shelf off the PRE. However, the extrapolation from the pore water data on the inner shelf to a scenario on the outer shelf where no pore water data are available is questionable. Even on the inner shelf, although in Core PR3 sands and gravels are the main components in the lower sediments, salinity is greater than 10 at these depths (Figure S3). So is the salinity at the lower sediment in Core HK9. Thus, the extrapolation that the authors made in assuming that sediments at depths where sands were present on the outer shelf have freshened groundwater and no more saline mixing is kind of speculative and needs further validation. The estimation of the volume of freshened groundwater based on such extrapolation, therefore, has large uncertainty, which needs to be assessed.

RESPONSE: Accepted.

Thanks for your comment, which can be summarized into 2 points: (1) the extrapolation from porewater data on the inner shelf to the outer shelf where no porewater data are available is questionable and needs further validation; (2) Even on the inner shelf, although in Core PR3 sands and gravels are the main components in the lower sediments, salinity is greater than 10 at these depths.

Reply to point (1): To address this point, we have made a great effort to collect nine more boreholes (GK04, KP103, ZHU1, ZHU2, ZHU3, PY27, P30, LH1-1, and DLW3101) in the outer shelf from Chinese National Offshore Oil Company, South China Sea Institute of Oceanology, Chinese Academy of Sciences, and Guangzhou Marine Geological Survey, which were added to Fig. 2 (attached here for easy reference) in the revised manuscript. All boreholes can provide more stratigraphical information to characterize the spatial distribution of the offshore aquifer system, showing that the offshore aquifers are really areally extensive. Geological transects based on these boreholes also prove that the offshore aquifers in the subaqueous delta on the outer shelf are connected with the nearshore aquifer system.

Fig. 2 Reconstructed Quaternary isopach map and paleochannels (a) and measured salinity profiles (b).

Major comments from Reviewer #1

To further address this comment, we have also constructed a set of sophisticated paleo-hydrogeological models considering marine regression and transgression during the 125 kyr BP to extrapolate the potential OFG volume in the outer shelf similar to other studies^{1,2,3,4} to calculate the OFG volume (Fig. 4 in the revised manuscript). The modelling results further confirmed that OFG can extend all the way to the outer shelf. Compared to the previous publications^{1,2,3,4}, which were largely based on numerical models with data from only a few offshore boreholes, our paper is based on the most comprehensive observed salinity data from an extensive set of offshore boreholes. (Lines 223-269)

Furthermore, we also used different hydraulic conductivities for the key hydrogeological units to evaluate the sensitivity of the OFG volume to the stratigraphic connectivity and provided a statistical result of the OFG volume (Table 1) (Line 247)

Table. 1 The volume of OFG in the PRE and adjacent continental shelf estimated by numerical model.

Scenarios	Base case	Case c	Case d	Case e	Case f	Case g	Case h
V_{TS} (km ³)	586.2	597.4	472.5	605.2	572.5	575.5	620.3

Fig. 4 Hydrogeological model results with different scenarios for the cross-section A-B and isotopic compositions of porewater with different endmembers. (Details can be found in revised manuscript)

Reply to point (2): We understand that there are two sites (PR3 and HK9-10) near Hong Kong with salinity greater than 15. These may be caused by geological heterogeneity or human activities. This area has lots of offshore engineering projects⁵, including cross-sea bridges and tunnels, navigation channels, sand dredging, and large-scale reclamation projects, which may create some contamination to the offshore

	aquifer. Fortunately, this relatively high salinity OFG seems to occur locally and be isolated since other boreholes (i.e., EH1 and HK10) around these two holes have salinity much lower. For example, EH1 has salinity as lower as 6.2 and the basal aquifer of HK10 has salinity of about 5. (Lines 145-152) Reply to Comment (b): we have deleted the practical salinity unit PSU in the revised manuscript.  1. Post, V. E., Groen J., Kooi H., Person M., Ge S., & Edmunds W. M. (2013), Offshore fresh groundwater reserves as a global phenomenon, Nature, 504(7478), 71-78. 2. Micallef, A., Person M., Haroon A., Weymer B. A., Jegen M., Schwalenberg K., Faghih Z., Duan S., Cohen D., Mountjoy J. J., Woelz S., Gable C. W. (2020), 3D characterisation and quantification of an offshore freshened groundwater system in the Canterbury Bight, Nature Communications, 11(1), 1372. 3. Cohen, D., Person M., Wang P., Gable C. W., Hutchinson D., Marksamer A., Dugan B., Kooi H., Groen K., Lizarralde D., Evans R. L., Day-Lewis F. D., & Lane J. W. (2010), Origin and extent of fresh paleowaters on the Atlantic continental shelf, USA, Ground Water, 48(1), 143-158 4. Person, M., Wilson J. L., Morrow N., & Post V. E. A. (2017), Continental-shelf freshwater water resources and improved oil recovery by low-salinity waterflooding, AAPG Bulletin, 101(01), 1-18, 5. Jiao, J., & Post V. (2019), Coastal Hydrogeology, Cambridge University Press, Cambridge.
Minor comments from Reviewer #1	Comment 1: Line 36: “dynamics” is ambiguous in its meaning here. Not sure what “freshwater source dynamics” means here. RESPONSE: Accepted. We have deleted the word “dynamics” and rewritten the sentence. (Lines 29-31)  Comment 2: Line 70: what is “the coastal zone”? Specify. RESPONSE: Accepted. We provided more specific information instead of “the coastal zone” here. (Lines 58-59)  Comment 3: Line 109: change “located” to “is located” RESPONSE: Accepted. Correction has been made as suggested. (Line 95)  Comment 4: Line 111: change “the three” to “three” RESPONSE: Accepted. Correction has been made as suggested. (Line 96)  Comment 5: Line 126: change “is” to “are”. RESPONSE: Accepted. We sorry for the spelling mistake, correction has been made as suggested. (Line 112)  Comment 6: Line 128: change “is” to “are”. RESPONSE: Accepted. Correction has been made as suggested. (Line 114)

Minor comments from Reviewer #1	Comment 7: Line 129: change “are” to “can”, remove “can”. RESPONSE: Accepted. Correction has been made as suggested. We change “are” to “can” here. (Line 116)
	Comment 8: Line 141: change “in the continental shelf” to “on the continental shelf” RESPONSE: Accepted. Correction has been made as suggested. (Line 135)
	Comment 9: Line 142: change “South China” to “southern China”. RESPONSE: Accepted. Correction has been made as suggested. (Line 136)
	Comment 10: Line 154: “a unitless quantity equivalent to g/kg” is misleading since absolute salinity is in g/kg, but practical salinity is not equivalent to absolute salinity. RESPONSE: Accepted. To avoid misunderstanding, we have rewritten those sentences. (Line 140-141)
	Comment 11: Figure 2: subplots BH5 and HK10 should have x axis labels and titles; subplots PR1 and BH5 should have y axis titles. RESPONSE: Accepted. Thanks for your suggestion. We have added such information in subplots of Fig. 2. According to the comments from Reviewer 2#, water depth lines are also added in this figure.
	Comment 12: Line 189: change “in” to “on”. RESPONSE: Accepted. As suggested, correction has been made as suggested. (Line 180)
	Comment 13: Line 198: change “regard” to “regarded”. RESPONSE: Accepted. Thanks for your comment, correction has been made here as suggested. (Line 197)
	Comment 14: Line 214-216: how did the authors jump from the calculated total static volume of the OFG of $61.7 \times 10^9 \text{ km}^3$ to the potential static OFG volume of $523.3 \times 10^9 \text{ km}^3$? It is confusing that the value of the potential static OFG volume just appeared without any calculation or explanation. Why did the two volumes differ? What is the purpose of presenting Eq. (1) here since the total volume is not what the authors need? RESPONSE: Accepted. Thanks for your comments on the manuscript, we have rewritten those sentences to address your concerns and hope that it is now clearer. (Section 2.3, Lines 200-269)

Minor comments from Reviewer #1	Here we calculated two potential static volume of OFG. The volume of 61.7 km³ is calculated for the inner shelf (using the ZK3, ZK4, PR1, BH1-13 as the boundary, roughly along the water depth of 40 m), where many boreholes with abundant porewater data and marine seismic reflections are available. Therefore, the OFG volume can be calculated by 3D geological interpolation using geostatistical tools. This volume can be regarded as the minimum volume of the OFG. We believe that OFG must exist beyond the inner self and the total volume of OFG must be much greater than 61.7 km³ because the salinity of porewater in boreholes ZK3, ZK4, PR1, and BH1-13 in the margin of the inner shelf is only at 1.0~9.0. Besides, all boreholes in the outer shelf can provide more stratigraphical information to characterize the spatial distribution of the offshore aquifer system that stores the OFG. Geological transects based on these boreholes (Fig. S5, supplementary materials) also prove that the offshore aquifers in the subaqueous delta on the outer shelf are connected with the nearshore aquifers. The potential static OFG volume of 523.3 km³ in the whole subaqueous delta of the Pearl River is estimated by extrapolation from the porewater data on the inner shelf to the outer shelf. This volume is less certain compared to estimation of OFG in the inner shelf due to less observed porewater data. Therefore, a set of sophisticated paleo-hydrogeological model considering marine regression and transgression during the Quaternary period (~125 kyr) were added in the revised manuscript to enhance the estimation. In summary, we concluded that the OFG has a volume of 546.4 ± 73.9 km³ (523.3 km³ in the previous manuscript). We introduced the calculation method of OFG volume in the revised manuscript, so it is unnecessary to put this simple formula along here, we have deleted it in the revised manuscript.
---	---

----- End of response to **Reviewer 1#**

General comments from Reviewer #2	Comment 1: This is an excellent manuscript worthy of publication in Nature Communication. This is the first study to document the existence of extensive vast (> 500 km³) offshore fresh to brackish water resources at shallow depths (50 m below the sediment water interface) in the South China Sea in an active delta system. It is highly likely that these freshwater to brackish water resources extend to greater depths. The authors argue that rapidly avulsing deltas will sequester vast amounts of fresh to brackish water found in sub-aqueous deltas during Pleistocene Sea level low stands. I do have a number of suggested revisions that I believe would improve the manuscript. RESPONSE: We would like to thank you for the positive comments and valuable suggestions on our manuscript. We have carefully addressed your comments in the revised manuscript and gave our point-to-point reply to your comments as presented below.
Specific comments from Reviewer #2	Comment 2: In general, I found the attempts to portray the three-dimensional topography as not very helpful (e.g., Fig 1b, 3b, c) given the large topographic range. The authors might consider plane-view plots instead. It would be helpful to see contours of the shallow (0-100m) bathymetry where the offshore wells are located. RESPONSE: Well taken. Thanks for your suggestion. We have used the plane-view plot to replace the original Fig. 1b to portray the topography of the South China Sea. But for Figs. 3b & c, we found that the 3D plots can provide a better visual representation of the data (OFG bodies and basement morphological distribution) and that the plane-view plots can present the data more directly, so we would like to keep them unchanged. But we also provided the plane-view for the basement in Fig. S4 in the supplementary materials. Comment 3: Along these lines, not much attention to well water depth is given in this manuscript. It is difficult to see if there is any correlation between seawater depth and salinity as suggested by Fig. 9 in Person et al. (2017). In general, shallower seawater depths areas would have been exposed longer to meteoric inputs of recharge. RESPONSE: Accepted. Thanks for your comments. The relationship between OFG and seawater depth is close as mentioned in Fig. 9 in the paper by Person et al. (2017), which studied five passive-margin cross sections and found that OFG is largely stored in the continental shelves within a water depth of 50 m. Our research in the PRE also supports this conclusion. According to Fig. 1d, all low-salinity boreholes are located within a water depth of 50 m. This important reference by Person et al. (2017) is now cited our manuscript. In fact, we have added the bathymetry data in Fig. 1d in the manuscript, but in order to emphasize this relationship further, we also added the water depth isoline in Fig. 2 in the manuscript. 1. Person, M., Wilson J. L., Morrow N., & Post V. E. A. (2017), Continental-shelf freshwater water resources and improved oil recovery by low-salinity waterflooding, AAPG Bulletin, 101(01), 1-18, doi: 10.1306/05241615143.

Specific comments from Reviewer #2

Comment 4: Figure 2 presents the select salinity profiles (i.e., the freshest wells). There are others that are more salty. All salinity profiles should be included. Perhaps, if necessary, in supplemental materials.

RESPONSE: Accepted.

Thanks for your suggestion. There are a total of 31 boreholes with porewater data. In addition to the 7 boreholes presented in Fig. 2, all the rest of the boreholes with salinity profiles are included in Fig. S2 in the supplementary materials. Note that we only managed to collect limited samples for porewater chemistry from the offshore boreholes BH1-13 have only because these boreholes were drilled for sea sand exploration by other projects, not specifically for our research. (Fig. S2, Lines 37-41)

Comment 5: Not much attention in the manuscript is paid to river avulsion and its potential roll in preserving offshore freshwater (schematic, Fig. 4). Active deltas have a lot of channel switching...see for example, Fig. 13 from Allison et al. (2003). Could there be a correlation between groundwater salinity and the age of a given delta front?

RESPONSE: Accepted.

Thanks for your comments. We have added such information in the schematic diagram in the manuscript (when you said Fig. 4, we guess you meant Fig. 5). We agree that the river avulsion has a potential role in preserving OFG. However, different from the Mississippi River Delta and Ganges-Brahmaputra delta, which are fairly flat¹ (see left figure below), the Pearl River Delta is constrained by lots of uplands and highlands of igneous rocks, so the river avulsion is less significant² (see right figure below). Our quick inspection shows there is not really an obvious correlation between groundwater salinity and the age of the current subaerial delta front (see right figure below). However, we do appreciate your comments and will see if there is such a relationship in the sub aqueous delta when more data are available.

Figure Comparison of the evolutions of the Ganges-Brahmaputra delta (a) and the Pearl River delta (b).

- Allison, M. A., Khan S. R., Goodbred S. L., & Kuehl S. A. (2003), Stratigraphic evolution of the late Holocene Ganges-Brahmaputra lower delta plain, *Sedimentary Geology*, 155(3), 317-342, doi: [https://doi.org/10.1016/S0037-0738\(02\)00185-9](https://doi.org/10.1016/S0037-0738(02)00185-9).
- Kuang, X., Jiao J. J., & Wang Y. (2016), Chloride as tracer of solute transport in the aquifer-aquitard system in the Pearl River Delta, China, *Hydrogeology Journal*, 24(5), 1121-1132.

Specific comments from Reviewer #2	Comment 6: The article Zhang et al. (2011) is not accessible to English readers, so it is hard to evaluate this citation. That is a pity. RESPONSE: Well taken. Thanks for your comments. It is not in English, but the article has an English abstract and is available online. It was published in a peer-reviewed reputable Chinese journal. We found that this article was also cited in Post et al. (2013), which have translated the key information in English in Table 1 in their paper. In fact, the story about the discovery of the offshore fresh groundwater was widely reported in Chinese news media and we trust that the information is very reliable. 1. Post, V. E., Groen J., Kooi H., Person M., Ge S., & Edmunds W. M. (2013), Offshore fresh groundwater reserves as a global phenomenon, Nature, 504(7478), 71-78, doi: 10.1038/nature12858.
	Comment 7: Line 104-105 The Zamrsky calculations of offshore freshwater are almost certainly an overestimate. These authors used unrealistically optimistic representations of continental shelf hydrogeology. RESPONSE: Well taken. We totally agree with the reviewer. The Zamrsky calculations relying solely on the numerical simulation based on over-simplified continental shelf hydrogeological structure have great uncertainty. Compared to these hypothetical numerical simulations, our estimation of OFG, which is based on so far, the most extensive and solid offshore dataset (= 31 offshore boreholes, > 2000 porewater geochemical data, and > 5000 km seismic profiles) is much more reliable.
	Comment 8: In the discussion section, the authors argue the utility of plots of stable isotopes. It would be nice to see a plot of $\delta^{18}\text{O}$ versus chloride concentrations. Yet, I don't see this plotted for the 31 offshore wells presented in this study. So, I don't find Figure 4 particularly important. RESPONSE: Accepted. The relations between $\delta^{18}\text{O}$ and $\delta^2\text{H}$ and Cl of the porewater are important because they can be used as indirect constraints for the origin of OFG (i.e., meteoric water, gas hydrate dissociation, and clay mineral dehydration). We presented three plots of $\delta^{18}\text{O}$ versus chloride for PR1, PR2, and ODP 1146 and they all lead to the same conclusion that the OFG is not from gas hydrate dissociation or clay mineral dehydration, but meteoric water. We did not analyse $\delta^{18}\text{O}$ for all the porewater samples from 31 offshore wells, which we think is unnecessary. Considering the reviewer's suggestion, we combined all the data from the three holes to form one plots of $\delta^{18}\text{O}$ versus chloride to make the paper more concise (Figs. 4 i, j in the revised manuscript).
	Comment 9: The authors seem to be arguing here that active delta systems are excellent venue for the sequestration of offshore freshened groundwater. Early discoveries of offshore freshwater were found along passive margins with relatively slow rates of riverine inputs of sediments and surface water (Hathaway et al. 1979). Interestingly, much of the freshwater is sequestered in pre-Pleistocene

Specific comments from Reviewer #2	sediments (Meisler et al. 1984) at depths of about 200 m. Here, the freshwater is found in shallow, young sediments. It might be interesting to compare the volumes of freshwater stored in the Pearl River delta and the USA North Atlantic Shelf. RESPONSE: Accepted. Yes, it has been well known that OFG is distributed along passive margins with relatively slow rates of riverine inputs of sediments and sequestered in pre-Pleistocene sediments beneath the Atlantic continental shelf of North and South America. Our paper discovered, with the support of very extensive solid data set, that the freshwater is also distributed in shallow, young sediments. We believe this phenomenon exists not only in Pearl River subaqueous delta but also in other subaqueous deltas at least in Red River, Mekong River, Yangtze River, and Yellow River etc in East and South China Sea. The thickness of the Quaternary strata in these rivers can be up to 1500 m¹, so the OFG there should be much more abundant than Pearl River subaqueous delta that has Quaternary strata of about 300-400 m. As suggested, we also estimated the total volumes of OFG stored in the PRE and adjacent continental shelf to be 4.7 km³ km⁻¹ using our paleo hydrogeological model, which is greater or much greater than that calculated in previous publications (3.24 -4.78 km³ km⁻¹ offshore of Canterbury, 1.6-1.8 km³ km⁻¹ offshore of New England, 4.4 km³ km⁻¹ offshore of New Jersey, 1.0 km³ km⁻¹ offshore of Jakarta, and 3.1 km³ km⁻¹ offshore Gippsland)². This comparison was added in the revised manuscript (Lines 265-269).  1. Clift, P. D., & Sun Z. (2006), The sedimentary and tectonic evolution of the Yinggehai-Song Hong basin and the southern Hainan margin, South China Sea: Implications for Tibetan uplift and monsoon intensification, Journal of Geophysical Research: Solid Earth, 111(B6), 2. Cohen, D., Person M., Wang P., Gable C. W., Hutchinson D., Marksamer A., Dugan B., Kooi H., Groen K., Lizarralde D., Evans R. L., Day-Lewis F. D., & Lane J. W. (2010), Origin and extent of fresh paleowaters on the Atlantic continental shelf, USA, Ground Water, 48(1), 143-158,
	Comment 10: Are there correlations between distance from a paleo-channel and the salinity of offshore wells? It doesn't look so obvious from Fig. 2. RESPONSE: Well taken. Interesting point! You are right, there is not a clear correlation. We will re-examine this point when we accumulate more offshore borehole data.
	Comment 11: What were the fluid pressure conditions during drilling? That is, were there artesian conditions? There might not be due to the large number of unmapped paleo-channels. RESPONSE: Well taken. Again an interesting point. A pumping test was conducted in the offshore aquifer about 60 m below the seabed near the east artificial island reclaimed for building the Hong Kong-Zhuhai-Macau Bridge, it was found that the hydraulic head of the aquifer was 0.53 m above the sea level with a salinity of 6.7 psu¹. Therefore, artesian conditions do exist in the offshore aquifers. This also suggests that the OFG is highly likely to be recharged by water from onshore highlands. However, we are not aware of any information on artesian conditions at other offshore drilling sites. The main goal of the drilling was to understand the geology and collect core samples so detailed pressure information was not recorded.

Specific comments from Reviewer #2	Comment 12: Line 101: change “since” - “during”. RESPONSE: Accepted. Correction has been made here as you suggested. (Line 87)
	Comment 13: Line 106: change “still clues” - “suggests”. RESPONSE: Accepted. Correction has been made here as you suggested. (Line 92)
	Comment 14: Line 107: delete “enormous”. RESPONSE: Accepted. We have deleted the “enormous” here. (Line 93)
	Comment 15: Line 109: delete “representative”. RESPONSE: Accepted. We have deleted the “representative” here. (Line 94)
	Comment 16: The font in Fig. 1a is really small. Suggest making Fig 1b plane view. RESPONSE: Accepted. We have enlarged the font in Fig. 1. Besides, Fig. 1b has been changed in a plane view in the revised manuscript. More information can be found in the response to the General Comment of Reviewer #1.
	Comment 17: What would a marine electromagnetic field campaign (e.g., Gustafson et al. 2019; Attais et al. 2022) reveal in this region? RESPONSE: Well taken. Thanks for your suggestion, we noticed marine controlled-source electromagnetic (CSEM) methods have been successfully applied in New Zealand ¹, Hawaii Island ², the North Atlantic continental margin ³, and Malta ⁴. We also intend to apply CSEM methods to the Pearl River Estuary and adjacent continental shelf in the near future. By calibrating with offshore boreholes and porewater data, CSEM can provide more areally extensive information on the distribution of OFG.  1. Micallef, A., Person M., Haroon A., Weymer B. A., Jegen M., Schwalenberg K., Faghih Z., Duan S., Cohen D., Mountjoy J. J., Woelz S., Gable C. W., Averages T., & Kumar Tiwari A. (2020), 3D characterisation and quantification of an offshore freshened groundwater system in the Canterbury Bight, Nature Communications, 11(1), 1372, doi: 10.1038/s41467-020-14770-7. 2. Attias, E., Thomas D., Sherman D., Ismail K., & Constable S. (2020), Marine electrical imaging reveals novel freshwater transport mechanism in Hawai'i, Science Advances, 6(48), eabd4866, doi: 10.1126/sciadv.abd4866. 3. Bertoni, C., Lofi J., Micallef A., & Moe H. (2020), Seismic reflection methods in offshore groundwater research, Geosciences, 10(8), doi: 10.3390/geosciences10080299.

----- End of response to **Reviewer 2#**

General comments from Reviewer #3	Comment 1: 1 The study represents an interesting study about offshore freshened groundwater in the Pearl River Estuary. The authors show various data sets, including seismic data (with different resolutions), borehole data, and hydro-geochemical data. 2 The case study is very interesting, but in my opinion a better description of the available data is needed. RESPONSE: We would like to thank you for the positive comments and valuable suggestions on our manuscript. OFG is a topic of interest to both hydrogeologists and marine geologists, and listed as one of the strategic objectives of the IODP 2050 science framework. To make the manuscript more readable, we have carefully addressed your comments and given our point-to-point reply to your comments as presented below.
	Comment 2: In Section 3, the authors report an estimate of the potential static volume of OFG. In my opinion, it is not clear how many sand layers they have considered. To estimate the total volume, they assigned a porosity value to each layer. No information is provided on this. How have the authors chosen this parameter? What values have the authors adopted? From boreholes? It is important to add this data. An estimate of the error should be given. RESPONSE: Accepted. Two areally extensive sand layers are mainly considered in the estimation of OFG (Fig. S5 in the supplementary materials). More detailed information about how the OFG was estimated was discussed in the reply to Comment 14 of Reviewer 1#. These sand layers consist of fine sands, sands and even gravels and can be very coarse (See photos of the core samples in Fig. S2 in the supplementary) and obviously should have a large porosity of 0.3 or even larger. To address this comment, we collected 11 boreholes with detailed profiles of porosity data in both inner and outer selfies and summarised the porosity information in Fig. S6 in the supplementary materials. Based on these data, the average porosity of the fine sand and medium to coarse sand are set to 0.35 and 0.3 respectively in the estimation of the OFG. Such information is also added in the revised manuscript. (Lines 213-215, Figure S6 in the supplementary materials)
	Comment 3: The quaternary isopachic map was created by using seismic data because no borehole reached the base of the Quaternary sediments. I have some questions: what procedure has it been adopted to interpolate the data obtained from seismic lines? How have you translated the seismic section from time to depth? With a unique velocity profile? What kind of velocity? RMS velocity translated in terms of interval velocity? This is not clear from the text. RESPONSE: Accepted. The Quaternary geology is basic geology and has been extensively studied by various national organizations such as Guangzhou Marine Geology survey, Chinese National Offshore Oil Company, and South China Sea Institute of Oceanology, Chinese Academy of Sciences. The Quaternary isopach map (Fig. 2) was based on previous information from several very authentic Chinese and English publications^{1,2,3,4} and also new information from ~100 holes recently drilled (see Fig. S5 for locations).

General comments from Reviewer #3	In the first manuscript, we stated that “...most offshore boreholes with salinity profiles do not penetrate the entire Quaternary formation...”. This may be misleading because it seems that the reviewer thought all boreholes used in this study did not reach the base of the Quaternary sediments. The fact is: almost boreholes for the conventional geological survey or petroleum exploration, which were used for depicting the offshore 3D aquifer-aquitard system in this study, reached or even penetrated the base of the Quaternary sediments. The boreholes drilled for this study only focusing on porewater chemistry are, however, much shallower. To avoid the confusion, we modified the sentence into “most offshore boreholes drilled for hydrogeochemical profiles for this study (Fig. 2) do not penetrate the entire Quaternary formation”. (Lines 162-163).  Ren, Z., Yang, L., Li, C., Wang, W. (2010), The drawing and introduce to the distribution map of the Quaternary thickness in the northern South China Sea, South China Journal of Seismology, 30(3): 37-41. doi:10.13512/j.hndz.2010.03.001. (In Chinese with English abstract) Tang, C., Zhou D., Endler R., Lin J., & Harff J. (2010), Sedimentary development of the Pearl River Estuary based on seismic stratigraphy, Journal of Marine Systems, 82, S3-S16 Huang, Z., Zhang W., & Chai F. (1995), The submerged Pearl River (Zhujiang) Delta, Acta Geographica Sinica, 50(3), 206-214. Liu, H., Lin C., Zhang Z., Zhang B., Jiang J., Tian H., & Liu H. (2019), Quaternary sequence stratigraphic evolution of the Pearl River Mouth Basin and controlling factors over depositional systems, Marine Geology & Quaternary Geology, 39(1), 25-37. Comment 4: In addition, it is better to replace "quaternary isopachic map" with "quaternary isopach map". RESPONSE: Accepted. Correction has been made here as your suggestion. (Line 189) Comment 5: In Supplementary Material TS3, a basement is reported on seismic sections. Is it possible to provide also the interpretation of the base of Quaternary sediments? RESPONSE: Accepted. As suggested, we have provided the base of Quaternary sediments in Fig. S3 in the supplementary materials. The Quaternary strata is widely distributed in the northern South China Sea and delineated in the seismic reflections by continuous, high-amplitude and mid-strong reflections and labelled T20, which can be continuously tracked in the study area ¹. (Line 71 in the supplementary materials)  Li, G., Mei L., Ye Q., Pang X., & Li W. (2023), Post-rift faulting controlled by different geodynamics in the Pearl River Mouth Basin, northern South China Sea margin, Earth-Science Reviews, 237.
Specific comments from Reviewer #3	Comment 6: The authors should follow journal guidelines for formatting references. RESPONSE: Accepted. Thanks for your comment. All the references were reformatted. Comment 7: The title is too long (maximum 15 words). RESPONSE: Accepted. As suggested, the title was slightly changed to fit the limit of 15 words. (Lines 1-2)

Specific comments from Reviewer #3	Comment 8: Please, remove acronym from the abstract. RESPONSE: Accepted. We have removed all acronyms from the abstract as the suggested.
	Comment 9: Figure 2. I do not find the (a) in the text. Firstly, you should describe what the insets report. RESPONSE: Accepted. Thanks for your suggestion, we have added the missing figure label (a) in the revised manuscript.
	Comment 10: Figure 2. The point (b) should be report as (a) RESPONSE: Accepted. Correction has been made in the revised Fig. 2 in the revised manuscript. (Lines 174-175)
	Comment 11: Line 423. please, add in the main text the meaning SBS (sub-bottom seismic). You inserted just in supplementary text. RESPONSE: Accepted. As suggested, we have added the sub-bottom seismic (SBS) in the revised manuscript (Line 442)
	Comment 12: Line 431. "rate" RESPONSE: Accepted. Correction has been made as suggested in the revised manuscript. (Line 450)
	Comment 13: Supplementary materials. it is not clear how the time-depth conversion has been performed. RESPONSE: Accepted. Thanks for your comment, the missing information and references have been added in the supplementary materials ¹. $D_2 = 0.0002t^2 + 0.9549t - 154.01$ where D_2 is the depth below seafloor (m), t is two-way travel time (ms), and $R^2=0.9886$. 1. Li, W., Wang P., Zhang C., & Lu B. (2011), Researches on time-depth conversion of deep-seated basal strata of Pearl River Mouth basin, Chinese Journal of Geophysics, 54(2), 449-456.

----- End of response to Reviewer 3#

REVIEWERS' COMMENTS

Reviewer #1 (Remarks to the Author):

The authors have made changes to address the comments by the reviewers. For the revision, I have further comments as below.

The title is too broad considering the work was only on the Pearl River estuary and adjacent shelf.

"in a large-river deltaic estuary" is more appropriate.

Line 141: PSU is usually used for practical salinity unit. In addition, there is no need to define an abbreviation for the practical salinity as this abbreviation was applied only once. In the parenthesis it is better that the salinity of adjacent seawater is provided as a reference.

Line 147: "impacted" to "been impacted", "intensively" to "intensive"

Line 149: "to" to "into"

Line 215: use consistent significant numbers.

Line 238: according to Table 1, the maximum offshore distance is 250.6 km, which differs from the authors presented, 225 km. What did the authors intend to present? Rewrite to clarify.

Line 263: change "relaible" to "reliable". Is the volume estimate the average of the volume estimates from all the scenarios? If yes, say so. If no, explain what the number represents.

"reliable" is not proper here because a persistent increase in the volume of OFG with the hydraulic conductivity cannot support the statement that the volume estimate is reliable.

Line 265-268: the authors state that the volume of OFG in the PRE, 4.7 km^3 per km, is greater or much greater than in other areas. However, from the values provided, 4.7 is not greater than 3.24-4.78 and considering the uncertainty I doubt that 4.7 is still greater than 4.4.

Line 266: provide the error of the volume of OFG per km.

Line 477: I guess "reginal" should be "regional".

Reviewer #2 (Remarks to the Author):

I am satisfied with the revisions to this manuscript. The manuscript can be published as is.

Reviewer #3 (Remarks to the Author):

Dear Authors

Thank you very much. The study is a very interesting and well described. The study is multidisciplinary and show a interesting and wide dataset. Th results are very interesting.

Response to Comments by the Reviewers

Ref: “Offshore freshened groundwater in a large-river estuary and adjacent shelf as a significant water resource” (ID # NCOMMS-22-46287A) by Chong Sheng, Jiu Jimmy Jiao, et al.,

(We first quote the **Comments** and then give the **Responses**, the red line number refers to the ***Revised Manuscript with changes marked**)

- **Accepted** = we agree with the reviewer’s comment and have modified the text accordingly.
- **Well taken** = we partially agree with the reviewer’s comment and have partially modified the text.
- **Not agree** = we do not agree with the reviewer’s comment and provide an explanation and justification of our approach.

General comments from Reviewer #1	Comment 1: The authors have made changes to address the comments by the reviewers. For the revision, I have further comments as below. The title is too broad considering the work was only on the Pearl River estuary and adjacent shelf. “in a large-river deltaic estuary” is more appropriate. RESPONSE: Accepted. Thank you for your comments. Considering the journal’s word limit and editor’s suggestion on the title (no more than 15 words), we have modified the title as “Offshore freshened groundwater in a large-river estuary and adjacent shelf as a significant water resource”.
Specific comments from Reviewer #1	Comment 2: Line 141: PSU is usually used for practical salinity unit. In addition, there is no need to define an abbreviation for the practical salinity as this abbreviation was applied only once. In the parenthesis it is better that the salinity of adjacent seawater is provided as a reference. RESPONSE: Accepted. We have deleted the define of the abbreviation and provided an average salinity of adjacent bottom seawater in the following parenthesis. (Line 138)  Comment 3: Line 147: “impacted” to “been impacted”, “intensively” to “intensive”. RESPONSE: Accepted. Thank you for your comments. Correction has been made as suggested. (Line 144)  Comment 4: Line 149: “to” to “into”. RESPONSE: Accepted. Correction has been made as suggested. (Line 147)  Comment 5: Line 215: use consistent significant numbers. RESPONSE: Accepted. Correction has been made as suggested. (Line 207)

Comment 6: Line 238: according to Table 1, the maximum offshore distance is 250.6 km, which differs from the authors presented, 225 km. What did the authors intend to present? Rewrite to clarify. RESPONSE: Accepted. Correction has been made here. We wanted to present the maximum offshore distance 250.6 km estimated by the hydrogeological model here. The original offshore distance 225 ± 25 km is the median and error range simulated by the model; we have deleted this sentence. (Line 226)
Comment 7: Line 263: change “relaible” to “reliable”. Is the volume estimate the average of the volume estimates from all the scenarios? If yes, say so. If no, explain what the number represents. “reliable” is not proper here because a persistent increase in the volume of OFG with the hydraulic conductivity cannot support the statement that the volume estimate is reliable. RESPONSE: Accepted. Thanks for your comments. Yes, you are right, this volume is the average of volume estimates from all scenarios and the following “\pm value” is the standard deviation based on the values from Table 1. We have deleted the word “reliable” here and rewritten this sentence (Line 40)
Comment 8: Line 265-268: the authors state that the volume of OFG in the PRE, $4.7 \text{ km}^3 \text{ per km}$, is greater or much greater than in other areas. However, from the values provided, 4.7 is not greater than 3.24-4.78 and considering the uncertainty I doubt that 4.7 is still greater than 4.4. RESPONSE: Accepted. Thanks for your comment, we have rewritten these sentences and deleted this expression. “the average volume of OFG per km (V_{FT}) calculated by the hydrogeological model is equivalent to $4.7 \pm 0.44 \text{ km}^3 \text{ km}^{-1}$, while the volume of OFG in other passive continental margins mainly ranges between 1.0 and $4.8 \text{ km}^3 \text{ km}^{-1}$, i.e., $3.24\text{-}4.78 \text{ km}^3 \text{ km}^{-1}$ offshore of Canterbury ⁷, $1.7 \text{ km}^3 \text{ km}^{-1}$ offshore of New England ⁴², $4.4 \text{ km}^3 \text{ km}^{-1}$ offshore of New Jersey, $1.0 \text{ km}^3 \text{ km}^{-1}$ offshore of Jakarta, and $3.1 \text{ km}^3 \text{ km}^{-1}$ offshore of Gippsland ⁴.” (Lines 243-245)
Comment 9: Line 266: provide the error of the volume of OFG per km. RESPONSE: Accepted. We have provided the error of the volume of OFG per km here “4.7 ± 0.44”. (Lines 244)
Comment 10: Line 477: I guess “reginal” should be “regional”. RESPONSE: Accepted. Thanks for your comment, correction has been made here, “regional” is correct here. (Line 441)

General comments from Reviewer #2	Comment 1: I am satisfied with the revisions to this manuscript. The manuscript can be published as is. RESPONSE: We are delighted that the reviewer is satisfied with our revisions. Thank you for recommending our paper for publication in Nature Communication.
---	---

General comments from Reviewer #3	Comment 1: Dear Authors Thank you very much. The study is a very interesting and well described. The study is multidisciplinary and show a interesting and wide dataset. Th results are very interesting. RESPONSE: We are glad that the reviewer found our study interesting. Thank you for stating that our work is multidisciplinary and suitable for publication in Nature Communication.
---	---